# Δ133p53 isoform promotes tumour invasion and metastasis via interleukin-6 activation of JAK-STAT and RhoA-ROCK signalling

Hamish Campbell[1], Nicholas Fleming[2], Imogen Roth[2], Sunali Mehta[2,3], Anna Wiles[2,3], Gail Williams[2], Claire Vennin[4,5], Nikola Arsic[8], Ashleigh Parkin[4,5], Marina Pajic[4,5], Fran Munro[6], Les McNoe[7], Michael Black[7], John McCall[6], Tania L. Slatter[2], Paul Timpson[4,5], Roger Reddel [1], Pierre Roux[8], Cristin Print[3,9,10], Margaret A. Baird[2,3] & Antony W. Braithwaite[1,2,3]

Δ122p53 mice (a model of Δ133p53 isoform) are tumour-prone, have extensive inflammation and elevated serum IL-6. To investigate the role of IL-6 we crossed Δ122p53 mice with IL-6 null mice. Here we show that loss of IL-6 reduced JAK-STAT signalling, tumour incidence and metastasis. We also show that Δ122p53 activates RhoA-ROCK signalling leading to tumour cell invasion, which is IL-6-dependent and can be reduced by inhibition of JAK-STAT and RhoA-ROCK pathways. Similarly, we show that Δ133p53 activates these pathways, resulting in invasive and migratory phenotypes in colorectal cancer cells. Gene expression analysis of colorectal tumours showed enrichment of GPCR signalling associated with Δ133TP53 mRNA. Patients with elevated Δ133TP53 mRNA levels had a shorter disease-free survival. Our results suggest that Δ133p53 promotes tumour invasion by activation of the JAK-STAT and RhoA-ROCK pathways, and that patients whose tumours have high Δ133TP53 may benefit from therapies targeting these pathways.

---

[1] Children's Medical Research Institute, University of Sydney, Sydney, NSW 2145, Australia. [2] Department of Pathology, Dunedin School of Medicine, University of Otago, 56 Hanover Street, 9054 Dunedin, New Zealand. [3] Maurice Wilkins Centre for Molecular Biodiscovery, University of Otago, c/o The University of Auckland, Private Bag 92019, 1142 Auckland, New Zealand. [4] The Garvan Institute of Medical Research and The Kinghorn Cancer Centre, 370 Victoria St, Darlinghurst, NSW 2010, Australia. [5] Faculty of Medicine, St Vincent's Clinical School, 370 Victoria St, Darlinghurst, NSW 2010, Australia. [6] Department of Surgical Sciences, Dunedin School of Medicine, University of Otago, 201 Great King St, 9054 Dunedin, New Zealand. [7] Department of Biochemistry, School of Biomedical Sciences, University of Otago, 710 Cumberland St, 9054 Dunedin, New Zealand. [8] CNRS, Centre de Recherche de Biochimie Macromoléculaire de Montpellier, 1919 Route de Mende, 34293 Montpellier, France. [9] Department of Molecular Medicine and Pathology, Faculty of Medicine, The University of Auckland, Private Bag 92019, 1142 Auckland, New Zealand. [10] Bioinformatics Institute, University of Auckland, Private Bag 92019, 1142 Auckland, New Zealand. Hamish Campbell, Nicholas Fleming and Imogen Roth contributed equally to this work. Correspondence and requests for materials should be addressed to A.W.B. (email: antony.braithwaite@otago.ac.nz)

The *TP53* gene encodes 12 isoforms[1–5] that moderate p53 activities[1,6,7] as well as having independent functions. Several isoforms are dysregulated in human tumours leading to the suggestion that they promote tumour formation. For example, elevated levels of Δ40p53 are found in melanomas and in breast tumours, particularly the aggressive triple-negative cancer subtype[8,9]. Aberrant expression of Δ133p53 has also been reported in renal cell cancer[10], head and neck tumours[11], colon carcinoma[7], lung cancer[12] and again in breast cancer[8,13], where it is associated with increased risk of recurrence[13]. Thus, these isoforms appear to have oncogenic activities. In addition, transgenic mice modelling Δ133p53 (designated Δ122p53) are highly tumour-prone, display chronic inflammation and autoimmunity[14,15], and have elevated levels of pro-inflammatory cytokines, notably interleukin-6 (IL-6)[14,15]. Δ133p53 and Δ122p53 promote migration in scratch wound and Transwell assays, which are associated with changes in the polarity of actin fibres[16] and Δ122p53 promotes invasion of pancreatic tumour cells through a three-dimensional collagen matrix. IL-6 was found to be a key mediator of these migratory phenotypes[16].

In this paper, we test the importance of IL-6 on the oncogenic and inflammatory phenotypes of Δ133p53 by crossing Δ122p53 mice with *IL-6* null mice. We show that *IL-6* loss reduces tumour incidence and metastasis and that Δ122p53 and Δ133p53 upregulate JAK-STAT3 and RhoA-ROCK signalling pathways that contribute to cell migration and invasion. Thus, IL-6 is an important mediator of the oncogenic activity of the Δ133p53 isoform. We also show that human colorectal cancers with elevated *Δ133TP53* mRNA are more aggressive, being associated with a shorter disease-free survival.

## Results

**Loss of *IL-6* reduces JAK-STAT signalling**. We previously reported that Δ122p53 mice exhibit widespread inflammation accompanied by 1.5- to 2-fold elevated levels of several pro-inflammatory cytokines in the serum compared to *wt p53* (wild-type) mice, with the exception of IL-6, which was 12-fold higher[15]. These results suggest that IL-6 is an important contributor to the pro-inflammatory and oncogenic phenotypes in Δ122p53 mice. To test this, double-mutant Δ122p53/Δ122p53 (designated Δ122/Δ122) or Δ122p53/p53+ (designated Δ122/+) mice were generated by crossing Δ122p53 mice with *IL-6*-deficient mice[17]. IL-6 functions by activating the Janus kinases (JAKs), recruiting the signal transducer and activator of transcription (STAT)-3 to induce transcription of genes involved in inflammation[14]. Thus, mice deficient in IL-6 should show an overall reduction in downstream cytokine levels. The sera of 7- to 9-week-old Δ122/Δ122 IL-6+/+, Δ122/Δ122 IL-6−/− and *wt p53/ IL-6−/−* mice were thus used to quantitate the levels of cytokines and chemokines by Bioplex array. At this time, these mice show no pathology[15]. Results are shown in Fig. 1 and in Supplementary Fig. 1. A total of 20 molecules were assayed of which 17 molecules are in the JAK-STAT pathway. Of these, 12 were significantly lower in the serum of Δ122/Δ122 IL-6−/− mice ($p < 0.05$, one-tailed unpaired *t*-test), ranging from a 4-fold difference for Mcp-1 (Ccl-2), to 10-fold for interferon (IFN)-γ and 27-fold for IL-17. Sera from *wt p53/IL-6−/−* control mice generally showed similar levels of these molecules to Δ122/Δ122 IL-6+/+ mice or were lower (e.g., Tnf-α, Ccl-2 and IL-17 ($p < 0.05$, one-tailed unpaired *t*-test)) as we found previously[15]. Thus, the JAK-STAT3 pathway is active in Δ122p53 mice and appears to be largely regulated by IL-6.

**Loss of *IL-6* reduces tumour incidence and metastasis**. To determine if loss of IL-6 had an impact on tumour incidence,

mice were monitored for up to 600 days and sacrificed if tumours were observed or the mice were distressed. Detailed histological analysis showed that 26/28 (93%) Δ122/Δ122 IL-6+/+ mice (Fig. 2a), 8/9 (89%) of Δ122/Δ122 IL-6+/− mice (Fig. 2b) and 14/20 (70%) Δ122/Δ122 IL-6−/− mice (Fig. 2c) had developed malignant tumours at the time of death. The reduction in tumour incidence between Δ122/Δ122 IL-6+/+ and Δ122/Δ122 IL-6−/− mice was significant ($p = 0.036$, $\chi^2$-test, Table 1). The majority of Δ122/Δ122 IL-6+/+ mice had either T-cell lymphomas (64%, 18/28) or sarcomas (14%, 4/28), although 3 mice had co-existing malignancies, and 1 mouse had a localised B-cell lymphoma (Supplementary Fig. 2). Most tumours in Δ122/Δ122 IL-6+/− mice were either T-cell lymphomas (56%, 5/9) or sarcomas (22%, 2/9; Fig. 2b), however, 1 mouse had co-existing malignancies (Fig. 2b). For Δ122/Δ122 IL-6−/− mice the incidence of T-cell lymphomas decreased to 53% (10/19), and 21% (4/19) had sarcomas (Fig. 2c); and there were no co-existing tumours.

For Δ122/+ IL-6+/+ mice, 93% (26/28) had malignant tumours (Fig. 2d). Sarcomas were most common with 43% (12/28) followed by carcinomas at 14% (4/28). T-cell lymphomas were present in 32% of mice (9/28). For heterozygous Δ122/+ IL-6+/− mice (Fig. 2e), 50% (13/26) had sarcomas alone and 1 mouse had a carcinoma. In addition, 23% (6/26 mice) had lymphomas, of which 3 were T-cell and 3 were B-cell lymphomas (and of these, 1 animal with a T-cell lymphoma and 2 with B-cell lymphomas had co-existing malignancies). No tumours were evident in 23% (6/26) of mice. The predominant malignancies for the heterozygous Δ122/+ IL-6−/− mice were solid tumours with 35% (7/20) being sarcomas and 10% (2/20) carcinomas (Fig. 2f). In this group, only 1 T-cell and 3 B-cell lymphomas were detected. There was no evidence of malignancy in 35% of these mice (7/20), and there were no co-existing malignancies. Δ122/+ IL-6+/− and Δ122/+ IL-6−/− mice had significantly reduced tumour incidence compared to Δ122/+ IL-6+/+ mice ($p = 0.01$ and 0.015, respectively, $\chi^2$-test; Table 1).

Histological analyses showed that both lymphomas and sarcomas had spread to multiple organs, which varied with genotype. Histological examples for both tumour types are shown in Fig. 3a, b. Eighty-nine percent of the T-cell lymphomas from Δ122/Δ122 IL-6+/+ showed extensive spread to the liver, spleen, salivary gland, kidney and nodes (Fig. 3a). Of the 6 Δ122/Δ122 IL-6+/− mice with T-cell lymphoma, 4 had extensive spread, and of the 9 Δ122/Δ122 IL-6−/− mice with a T-cell lymphoma, 5 showed local spread and 4 mice had extensive spread. This was significantly lower compared to the Δ122/Δ122 IL-6+/+ mice ($p = 0.01$, $\chi^2$-test; Table 2). For sarcomas, 3/6 Δ122/Δ122 IL-6+/+ mice showed metastases to the gut, liver, spleen, pancreas or lymph node; 1/3 of Δ122/Δ122 IL-6+/− mice had a single metastasis to the heart; and 0/4 Δ122/Δ122 IL-6−/− mice had metastases, although this did not quite reach statistical significance ($p = 0.09$, $\chi^2$-test; Table 2) due to low numbers of tumours.

In Δ122/+ mice a similar trend was apparent. For Δ122/+ IL-6+/+ mice there was extensive lymphoma spread evident in 56% (5/9) of cases involving the lungs, liver, pancreas, gut, lymph nodes and other organs. In Δ122/+ IL-6+/− mice, 3/6 mice had T-cell lymphomas of which 2 displayed extensive spread. Of the Δ122/+ IL-6−/− mice, only 1 mouse had a T-cell lymphoma that showed limited spread. Despite the trend towards reduced tumour spread with loss of IL-6, there was no significant difference among the genotypes (Table 2). For sarcomas, 5/13 of Δ122/+ IL-6+/+ mice showed evidence of metastases involving multiple organs while only 1/16 Δ122/+ IL-6+/− mice showed evidence of metastases involving multiple organs (Table 2). This was statistically significant ($p = 0.03$, $\chi^2$-test; Table 2). There were 4/7 of Δ122/+ IL-6−/− mice with metastases but this was not significantly different from the Δ122/+ IL-6+/+ mice (Table 2).

However, the *Δ122/+ IL-6+/+* mice had more than one malignancy in the same animal, indicative of a more aggressive phenotype, which was not the case of mice without IL-6.

Previously, we also reported that *Δ122/Δ122* mice display splenomegaly prior to tumour onset indicative of cellular hyperproliferation[15]. To determine if this required IL-6 we measured the wet weights of the spleens in the mouse cohorts. Supplementary Fig. 3a shows the ratio of spleen weight to body weight of *Δ122/Δ122* mice (median ratio = 0.02, $p = 0.003$, two-tailed unpaired $t$-test) was reduced to the levels of *wt p53* mice by retention of 1 *wt p53* allele or the loss of 1 or 2 *IL-6* alleles (median ratio ~ 0.004). Similar results were obtained for liver (Supplementary Fig. 3b).

Despite the significant reduction in tumour incidence and metastasis in mice that had lost one or both *IL-6* alleles, the number of animals studied were not sufficient to ascertain a significant difference in survival between the genotypes, although

there was a trend towards increased survival for the heterozygous mice (Supplementary Fig. 4a and 4b). There was also no difference in the time at which tumours were detected (and mice culled) across the genotypes (Supplementary Fig. 5 and Supplementary Table 1). Of note, *p53+/+ IL-6−/−* mice did not develop tumours and were all culled after 500 days (Supplementary Fig. 4a) mainly due to a distended abdomen.

In summary, the loss of *IL-6* in *Δ122/Δ122* mice resulted in decreased levels of serum cytokines and chemokines involved in the JAK-STAT pathway and significantly decreased the incidence of T-cell lymphomas with extensive spread. There were also no co-malignancies and no metastatic sarcomas in these mice. Loss of *IL-6* in the *Δ122/+* mice resulted in a decreased incidence of T-cell lymphomas as well as a significant reduction in sarcomas that underwent metastasis. These results suggest that IL-6 is an important contributor to the pro-inflammatory and tumorigenic phenotypes of *Δ122p53* mice.

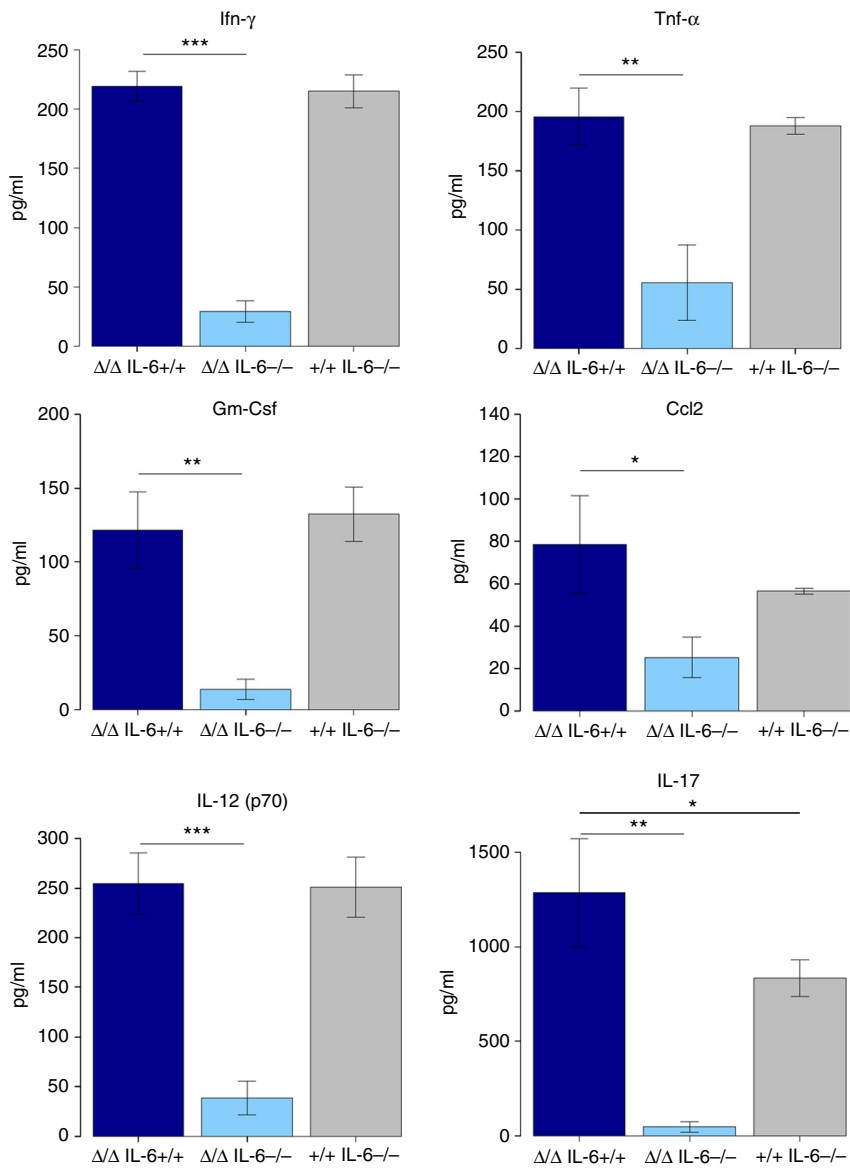

**Fig. 1** Reduced expression of pro-inflammatory cytokines in *IL-6*-deficient *Δ122p53* mice. Serum from *Δ122/Δ122 IL-6+/+* (n = 3), *Δ122/Δ122 IL-6−/−* (n = 3) and *+/+ IL-6−/−* (n = 3) mice was analysed using Bio-Plex Pro Mouse Cytokine 23-plex Array. Cytokine concentrations (pg/ml) are means ± s.e. m. Heteroscedastic one-tailed unpaired $t$-test was performed and $p < 0.05$ was considered significant. *$p < 0.05$, **$p < 0.005$ and ***$p < 0.0005$ are designated, respectively. Δ/Δ = *Δ122/Δ122*

**Δ122p53 activates RhoA-ROCK signalling**. As well as activating inflammatory gene transcription[20] via the JAK-STAT3 pathway, IL-6 also induces transcription of genes involved in migration[21], angiogenesis[22–24], metastasis[25] and tumour progression[26]. We previously demonstrated that Δ122p53 drives a migratory phenotype associated with actin polarisation[16]. To test whether this is reduced in the absence of *IL-6*, mouse embryonic fibroblasts (MEFs) were used in scratch wound closure assays and the proportion of cells showing actin polarisation was assessed. Examples of actin polarisation are shown in Fig. 4a and the quantitation is shown in Fig. 4b. Actin polarisation was reduced by 40–60% in the absence of one or both *IL-6* alleles (Fig. 4b). In parallel, as a control, we found that loss of *IL-6* reduced cell migration two- to four-fold in Transwell assays (Fig. 4c). These results suggest that IL-6 is required for actin polarisation induced by Δ122p53, a process driven by GTPase RhoA and associated kinase ROCK[19,27,28]. These results suggest that RhoA-ROCK signalling is active in Δ122p53-expressing cells and this is dependent on IL-6.

Next, we tested whether inhibiting the JAK-STAT and RhoA-ROCK pathways could reduce invasion caused by Δ122p53. To do this we used mouse pancreatic adenocarcinoma (PDAC) cells that had been stably transduced to express Δ122p53 (PDAC/

Δ122)[16] in invasion assays through collagen matrices[29–31]. Cells were also incubated with the Pan-JAK inhibitor P6[28], the ROCK inhibitors Y-27632 and H1152[28], and the RhoA inhibitor TAT-C3 (C3)[29]. As previously demonstrated[16], PDAC/Δ122 (Δ) cells showed a greater degree of invasion (2.4-fold) than PDAC/vector cells (Vo; Fig. 5a). The results also show that the three inhibitors reduced invasion of PDAC/Δ122 cells to varying degrees. Thus, at least one mechanism of inhibiting the Δ122p53-driven invasion is by inhibiting the JAK-STAT3/RhoA-ROCK pathway. Further to this point, the 'invading cells' were stained with an antibody to phosphorylated STAT3 (pSTAT3) as a marker of JAK-STAT activation. Results (Fig. 5b) show that ~2-fold more PDAC/Δ122 cells express pSTAT3 compared to PDAC/vector cells, which was abolished with the JAK, ROCK and RhoA inhibitors. In other experiments, we stained for phosphorylated myosin phosphatase target subunit 1 (pMYPT1) as a marker of ROCK activity and actin contractility[19]. pMYPT1 staining was at background levels in PDAC/vector cells but 6- to 7-fold elevated in PDAC/Δ122 cells (Fig. 5c), which was reduced 2- to 4-fold with the JAK, ROCK and RhoA inhibitors. The non-invading and whole population of PDAC/Δ122 cells were also treated with inhibitors and stained for pSTAT3 and pMYPT1 (Supplementary Fig. 6). Again, pSTAT3 was elevated in Δ122p53 cells, which was abolished with all inhibitors. For pMYPT1, the non-invading cells and the whole PDAC/Δ122 cell population also showed elevated staining compared to PDAC/vector cells, but the inhibitors had minimal effects. Y-27632 reduced staining in the non-invading cells but overall had no significant effect, whereas C3 treatment reduced staining of the whole population by 40% in PDAC/Δ122 cells. These results demonstrate that inhibition of JAK-STAT and RhoA-ROCK pathways reduces Δ122p53-induced invasion, a process requiring IL-6.

**RhoA-ROCK signalling is upregulated by Δ133p53 isoforms**. To confirm that the RhoA-ROCK signalling pathways is upregulated by human Δ133p53 isoforms as implied by the above data, HCT116 cells were transfected with the three Δ133p53 isoforms (Δ133p53α, Δ133p53β and Δ133p53γ) and allowed to invade through Matrigel, in the presence and absence of the Y-27632 ROCK inhibitor. All three isoforms invaded the Matrigel, which was blocked by Y-27632 (Fig. 6a). Recently, it was reported that the two homologous ROCK isoforms are not functionally equivalent[32,33]. To determine whether both ROCK isoforms have identical effects on the Δ133p53-dependent 'rounded' phenotype important for migration, we measured expression of ROCK1 and ROCK2 in transfected cells (Fig. 6b, d). Expression of ROCK1 was slightly increased in adherent cells, as compared to controls, but was decreased in the rounded, non-adherent cells (Fig. 6c for an example of rounded cells—green; Fig. 6b for quantitation of ROCK1 levels). However, ROCK2 expression level was high in all

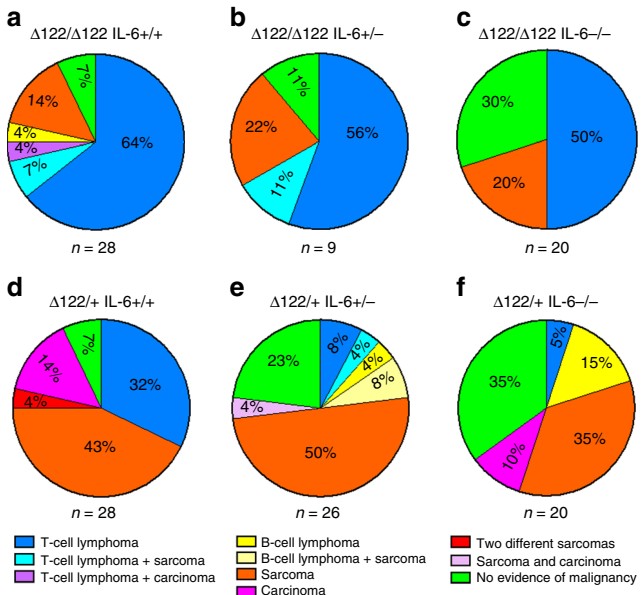

**Fig. 2** Loss of *IL-6* in Δ122p53 mice reduces tumour incidence. **a–f** Percentage of tumour types identified by histopathological analysis displayed as pie charts for each genotype. *n* = cohort size. All cohorts of mice were followed for up to 600 days and mice were culled when signs of disease or distressed were observed

---

**Table 1 Loss of IL-6 in Δ122p53 mice reduces tumour incidence**

| Comparison | Genotype | NEM | Tumours | $\chi^2$-statistic | *p*-value |
|---|---|---|---|---|---|
| Group 1 | Δ/Δ *IL-6+/+* | 2 | 26 | 0.144 | 0.704 |
| Group 2 | Δ/Δ *IL-6+/−* | 1 | 8 | | |
| Group 1 | Δ/Δ *IL-6+/+* | 2 | 26 | 4.39 | 0.036* |
| Group 2 | Δ/Δ *IL-6−/−* | 6 | 14 | | |
| Group 1 | Δ/+ *IL-6+/+* | 2 | 26 | 2.712 | 0.01* |
| Group 2 | Δ/+ *IL-6+/−* | 6 | 20 | | |
| Group 1 | Δ/+ *IL-6+/+* | 2 | 26 | 5.943 | 0.015* |
| Group 2 | Δ/+ *IL-6−/−* | 7 | 13 | | |

Summary of $\chi^2$-statistic and *p*-value for no evidence of malignancy (NEM) between the genotypes
*$p < 0.05$ was considered to be significant

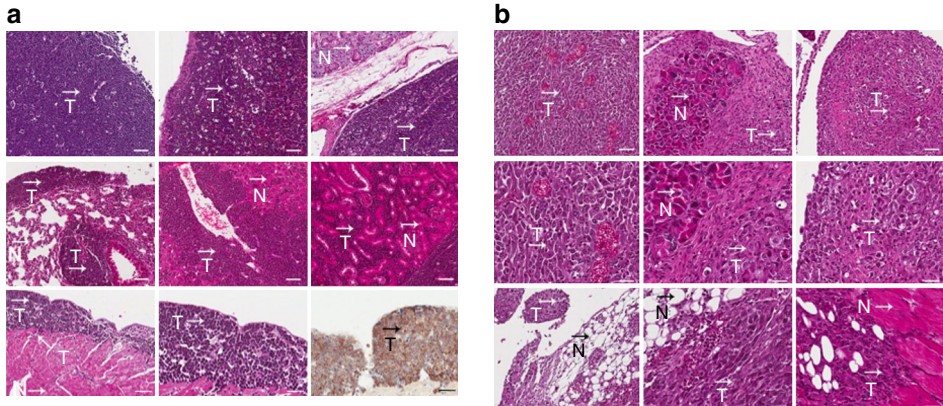

**Fig. 3** Loss of *IL-6* leads to reduced metastatic spread of tumours in Δ*122p53* mice. Necropsies were carried out on cohorts of mice from each genotype and histological examination carried out to ascertain tumour type and evidence of invasion. **a** Examples of photomicrographs illustrating histological (H&E) and immunohistochemical staining (chromogen-brown) of a T-cell lymphoma from a Δ*122/Δ122 IL-6+/+* mouse. From left to right, top to bottom, at ×100 magnification: primary T-cell lymphoma in the thymus; tumour spread to spleen, salivary gland node, lung, liver, kidney, heart, heart (×200); and CD3 staining (×200) of the heart. Scale bar = 50 μm (top and middle row), 50 μm (bottom—left) and 25 μm (bottom middle and bottom right). **b** Examples of photomicrographs illustrating histopathological (H&E) staining of a metastatic sarcoma from a Δ*122/Δ122 IL-6+/+* mouse. From left to right, top: primary sarcoma in the abdomen and metastases to pancreas and pancreatic node (×100); middle: the same at higher magnification (×200); bottom: metastases involving adipose tissue at ×100 and ×200 magnification, and metastases involving skeletal muscle (×200). Scale bar = 50 μm (top row), 25 μm (middle row) and 50 μm (bottom—left) and 25 μm (bottom middle and bottom right). T denotes tumour and N denotes normal tissue. ×100 magnification unless otherwise stated

**Table 2 Loss of *IL-6* leads to reduced metastatic spread of tumours in Δ122p53 mice**

| Comparison | Genotype | Local | Extensive | $\chi^2$-statistic | *p*-value |
|---|---|---|---|---|---|
| Group 1 | Δ/Δ IL-6+/+ | 2 | 17 | 1.77 | 0.18 |
| Group 2 | Δ/Δ IL-6+/− | 2 | 4 | | |
| Group 1 | Δ/Δ IL-6+/+ | 2 | 17 | 6.6 | 0.01* |
| Group 2 | Δ/Δ IL-6−/− | 5 | 4 | | |
| Group 1 | Δ/+ IL-6+/+ | 1 | 5 | 0.321 | 0.57 |
| Group 2 | Δ/+ IL-6+/− | 1 | 2 | | |
| Group 1 | Δ/+ IL-6+/+ | 1 | 5 | 0.19 | 0.66 |
| Group 2 | Δ/+ IL-6−/− | 0 | 1 | | |

| Comparison | Genotype | No metastasis | Metastasis | $\chi^2$-statistic | *p*-value |
|---|---|---|---|---|---|
| Group 1 | Δ/Δ IL-6+/+ | 3 | 3 | 0.225 | 0.64 |
| Group 2 | Δ/Δ IL-6+/− | 2 | 1 | | |
| Group 1 | Δ/Δ IL-6+/+ | 3 | 3 | 2.857 | 0.09 |
| Group 2 | Δ/Δ IL-6−/− | 4 | 0 | | |
| Group 1 | Δ/+ IL-6+/+ | 8 | 5 | 4.53 | 0.03* |
| Group 2 | Δ/+ IL-6+/− | 15 | 1 | | |
| Group 1 | Δ/+ IL-6+/+ | 8 | 5 | 0.642 | 0.42 |
| Group 2 | Δ/+ IL-6−/− | 3 | 4 | | |

Summary of $\chi^2$-statistic and *p*-value for local vs. extensive organ involvement by T lymphomas between the genotypes and number of metastasis of sarcomas between the genotypes
*$p < 0.05$ was considered to be significant

situations (Fig. 6d). This suggests that ROCK2 is preferentially required for the rounded cell movement. ROCK-driven actin reorganisation during motility is largely dependent on RhoA[27,34]. We therefore asked if Δ133p53 isoforms exert their effects on ROCK signalling via RhoA by testing whether the active GTP-bound form of RhoA is elevated in cells transfected with the isoforms. Control levels of RhoA were found in adherent HCT116 cells, but this was markedly increased in rounded cells, particularly those transfected with Δ133p53β, which showed 2–3 times more GTP-RhoA than cells transfected with Δ133p53α or Δ133p53γ (Fig. 6e).

**Elevated Δ133p53 is associated with higher cancer recurrence.** The above studies suggest that elevated levels of the

Δ133p53 isoform(s) may cause a more aggressive disease. To test this, we analysed 35, mainly stage II, human colorectal tumours (CRC) for Δ*133TP53* and *FLTP53* mRNA levels and compared these with patient disease-free survival. Colorectal tumours are known to be driven by inflammation[35] and importantly, mutant p53 has been demonstrated to sustain inflammation in colorectal cancer[36]. Our analysis showed that Δ*133TP53* and *TP53β* transcript variants were abundantly expressed in a subset of these tumours compared to the *FLTP53* transcript. Interestingly, the relative Δ*133TP53* transcripts correlated with *TP53β* transcript ($R^2 = 0.99$, $p = 0.0$, two-tailed *t*-test; Fig. 7a) but not with *FLTP53* ($R^2 = 0.06$, $p = 0.2$, two-tailed *t*-test; Fig. 7b). This suggests that the predominant Δ*133TP53* family transcript is Δ*133TP53β*. Our data also show that there was no significant difference in disease-

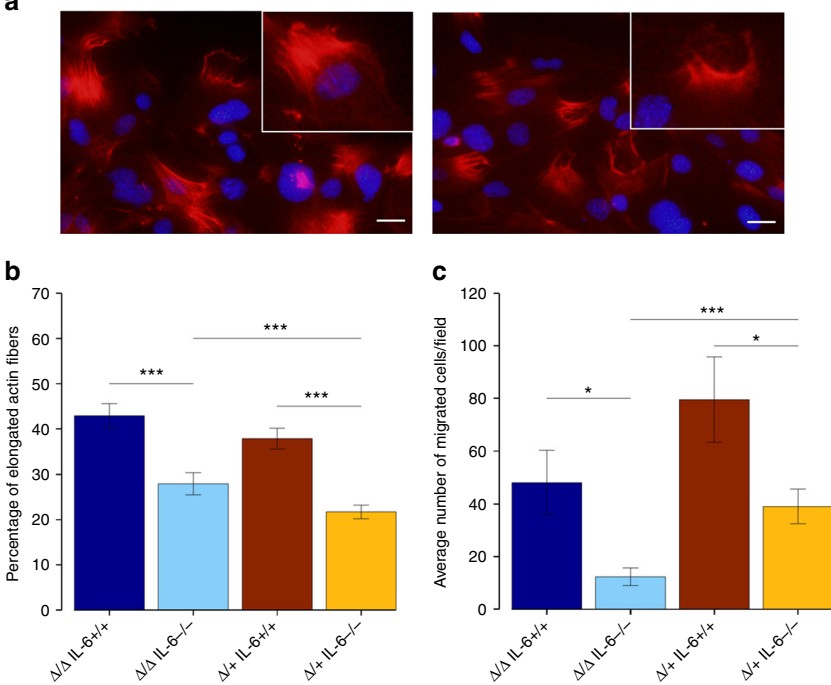

**Fig. 4** Loss of *IL-6* reduces cell migration and actin polarisation in Δ122p53-expressing MEFs. **a** Examples of phalloidin staining illustrating actin fibre polarisation in Δ122/Δ122 IL-6+/+ MEFs (images taken at ×400 magnification). Scale bar = 20 μm. Migrating cells were stained with Hoescht (blue) and Phalloidin (red). **b** The average % of phalloidin-stained cells with elongated actin fibres was quantitated. This was done with 'blinding' by two independent researchers. **c** The average number of migrated MEFs/field 4 h after seeding into Transwell inserts. The bars represent the mean and error bars represent ±s.e.m. of 15 independent fields. A *p*-value was obtained after performing a heteroscedastic, two-tailed unpaired *t*-test between different groups. *p* < 0.05 was considered significant. *\*p* < 0.05 and *\*\*\*p* < 0.0005 are designated, respectively

free survival of patients with tumours expressing the highest level of the *FLTP53* mRNA (Fig. 7c; *p* = 0.67, log rank test), but those expressing the highest levels of the *Δ133TP53* mRNA (>80th percentile) were associated with a significantly shorter disease-free survival (Fig. 7d; *p* = 0.046, log rank test). The tumours with elevated levels of *Δ133TP53* mRNA all had a wild-type *TP53* gene (based on sequencing exons 5–8).

In addition to tumour stage, decreased lymphocytic infiltration has been reported to be associated with poor prognosis for CRC patients[37,38]. Analysis of our cohort showed that tumours with low lymphocytic infiltration had significantly increased levels of *Δ133TP53β* mRNA expression (Fig. 7f; *p* = 0.037, two-tailed unpaired *t*-test), however, this was not the case for the *FLTP53* transcript (Fig. 7e; *p* = 0.37, two-tailed unpaired *t*-test). Moreover, lymphocytic infiltration alone failed to significantly discriminate patients with shorter disease-free survival (*p* = 0.452, log rank test).

Next, to identify pathways associated with *Δ133TP53* mRNA expression in CRCs we performed Spearman's correlation analyses combining relative *Δ133TP53* mRNA expression, determined by quantitative reverse transcription-PCR (RT-qPCR), with expression of genes from the Affymetrix Human Exon 1.0 ST arrays. We identified that *Δ133TP53* mRNA was associated with the expression of 364 genes (Spearman's correlation coefficient ρ ≥ 0.35; Supplementary Data 1). Of the GO biological processes over-represented by genes associated with *Δ133TP53* mRNA, leukocyte activation (GO:0045321 Bonferroni-corrected *p* = 0.015) and G-protein coupled receptor (GPCR) signalling pathway (GO:0007186 Bonferroni-corrected *p* = 0.01) were common. It is of note that GPCR signalling associated with *Δ133TP53* mRNA expression in these tumours is upstream of the JAK-STAT and RhoA-ROCK pathways.

## Discussion

Recent years have seen p53 implicated in many aspects of immune regulation; particularly those involved in the immune response to infection by viruses and other pathogens (reviewed in ref.[39]). Multiple Toll-like receptor genes are transcriptionally upregulated by p53[40], as well as several cytokine and chemokine genes, including *IFN-α*, *IFN-β* and *CCL2*[41,42]. In contrast, in other studies, p53 has been shown to inhibit STAT1, the transcription factor required to transactivate IFN-inducible genes and pro-inflammatory cytokines (reviewed in ref.[43]). p53 also represses transcription of *IL-6*, *IL-12* and *TNF-α* genes and other nuclear factor κB (NFκB)-regulated promoters (reviewed in ref.[44]), but it has also been reported to cooperate with NFκB[45]. p53 plays a direct role in the removal of damaged host cells in the innate immune response, by upregulating the *DD1α* gene required for clearance of apoptotic cells[46]. p53 upregulates the immune checkpoint molecules programmed cell death-1 (PD-1) and its ligand PDL-1[46] but p53 also upregulates mir-34, which binds the 3′-untranslated region of PDL-1 to downregulate expression[47]. In addition, p53 may directly affect the adaptive immune response as antigen-presenting (dendritic) cells derived from p53 null cells fail to elicit a cytotoxic T-cell response due to a failure to produce IL-12[48]. Thus, although p53 appears to regulate multiple components of the immune response, its roles are often contradictory.

One way perhaps to rationalise these differences is to consider the role of p53 isoforms. As outlined in the introduction, *Δ122p53* mice show profound inflammatory phenotypes and elevated cytokine levels[15] and MEF cells derived from these mice secrete higher amounts of IL-6 and the chemokines CCL2, CCL3 and CCL4 than cells with wt p53[16]. Furthermore, splenocytes from *Δ122p53* mice show increased transcription of genes in the

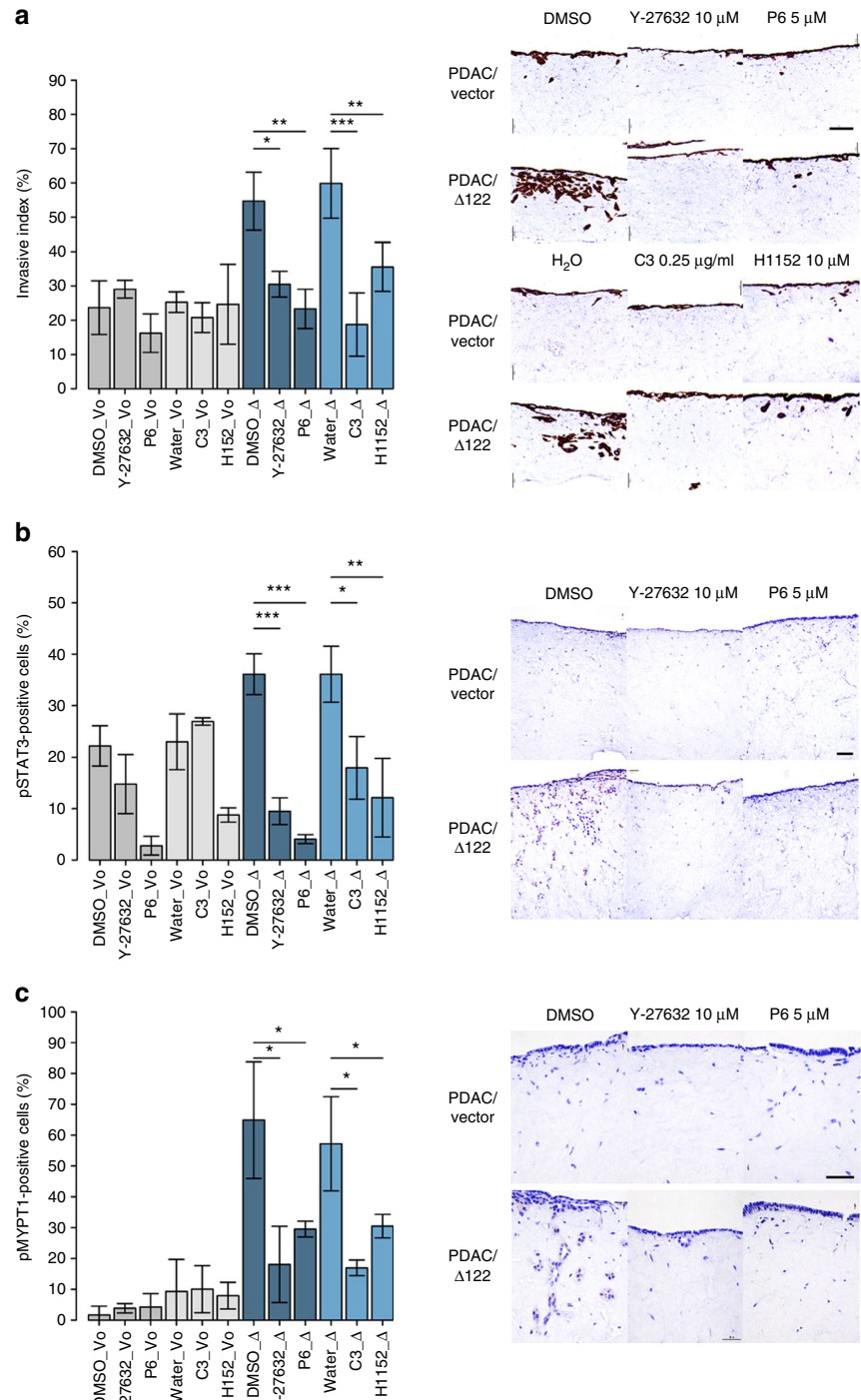

**Fig. 5** Δ122p53-expressing PDAC cells have constitutively active JAK-STAT and RhoA-ROCK signalling pathways. **a** Quantitation of invasive index and representative images of pan-cytokeratin s-stained sections of PDAC/vector (Vo) and PDAC/Δ122 (Δ) cells invading through collagen matrix contracted by TIFs treated with either control solvents, Pan-JAK inhibitor P6, the ROCK inhibitors Y-27632 and H1152, and the RhoA inhibitor TAT-C3, respectively. **b** Quantitation and representative images of pSTAT3-stained sections of invading PDAC cells upon treatment with the above inhibitors. **c** Quantitation and representative images of pMYPT1-stained sections of invading PDAC cells upon treatment with the above inhibitors. Scale bar = 100 μm. The bars represent the mean and error bars represent ±s.e.m of $n = 3$ repeats with 3 biological replicates each. p-values were determined using unpaired t-tests. $p < 0.05$ was considered significant. *$p < 0.05$, **$p < 0.005$ and ***$p < 0.0005$ are designated, respectively

IFN-γ pathway, and cells expressing human Δ133p53 isoform were found to regulate the *STAT1* gene[14]. In addition, infection of gastric epithelial cells with *Helicobacter pylori* activated the pro-inflammatory NFκB pathway, and this required the Δ133p53 isoform[49]. Thus, the positive effects of p53 on immune regulation, for example, in response to virus infection, might be attributed to the p53 isoform, whereas the repressive effects may be due to full-length p53. Depending on the activating signal one or other of these two p53 species will dominate. However, if there is sustained expression of the isoform as occurs in the Δ122p53 mice, then inflammatory pathologies may ensue.

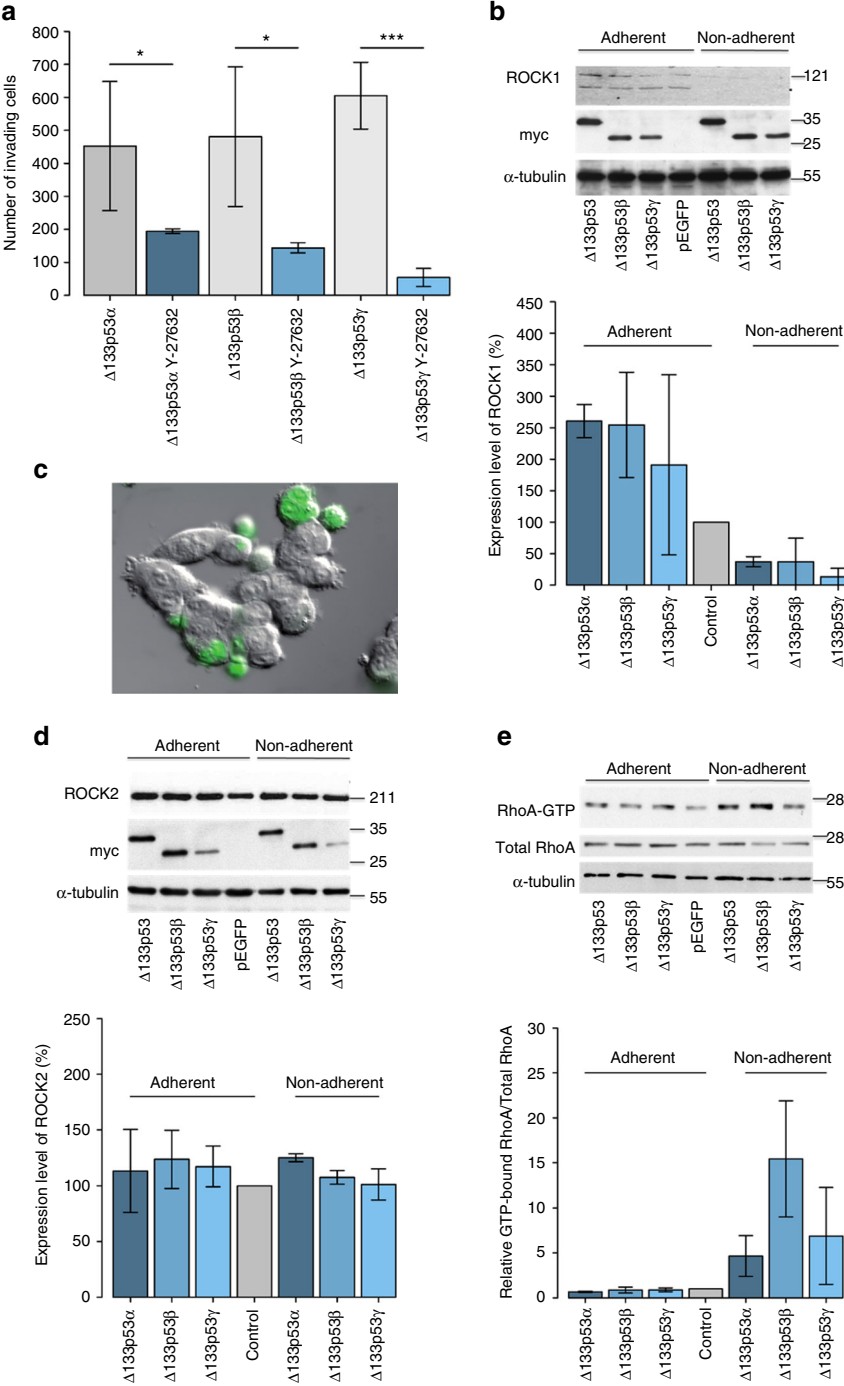

**Fig. 6** Invasiveness of HCT116 cells expressing Δ133p53 (α, β or γ) isoforms requires RhoA-ROCK activity. **a** Quantitative analysis of invasiveness of HCT116 cells expressing Δ133p53α, Δ133p53β or Δ133p53γ in the absence or presence of the ROCK inhibitor Y-27632. After 24 h transfection, cells were treated with 10 μM Y-27632 and 24 h later invasion assays were performed, with incorporation of Y-27632 into the Matrigel. Values are the mean number of invading cells ± s.d. (error bars); $n = 4$ independent experiments. **b** Western blot analysis of the expression of ROCK1 in HCT116 cells expressing the different constructs (control: myc alone; Δ133p53α: GFP or myc-tagged Δ133p53α isoform; Δ133p53β: GFP or myc-tagged Δ133p53β isoform; Δ133p53γ: GFP or myc-tagged Δ133p53γ isoform). Normalisation was performed using an anti-α-tubulin antibody and scanned autoradiographs were quantified. Values are the mean expression ratios ± s.d. (error bars); $n = 3$ independent experiments. **c** Differential interference contrast light microscopy and fluorescence microscopy of HCT116 cells expressing GFP-Δ133p53α showing their rounded appearance. Scale bars = 10 μm. The total number of transfected (GFP-positive) cells was 46 ± 5.8%, of which 12.6 ± 4.4% were non-adherent. **d** Western blot analysis of the expression of ROCK2 in HCT116 cells expressing the different constructs as in **b** and quantification carried out as in **b**. **e** RhoA activity in cells described in **b**. RhoA abundance was evaluated by western blotting. Results represented as the ratio of activated RhoA (GTP-bound RhoA) protein divided by total RhoA protein level. Values are the mean ± s.d. (error bars) of $n = 3$ independent experiments. $p < 0.05$ was considered significant. $*p < 0.05$ and $***p < 0.0005$ are designated, respectively

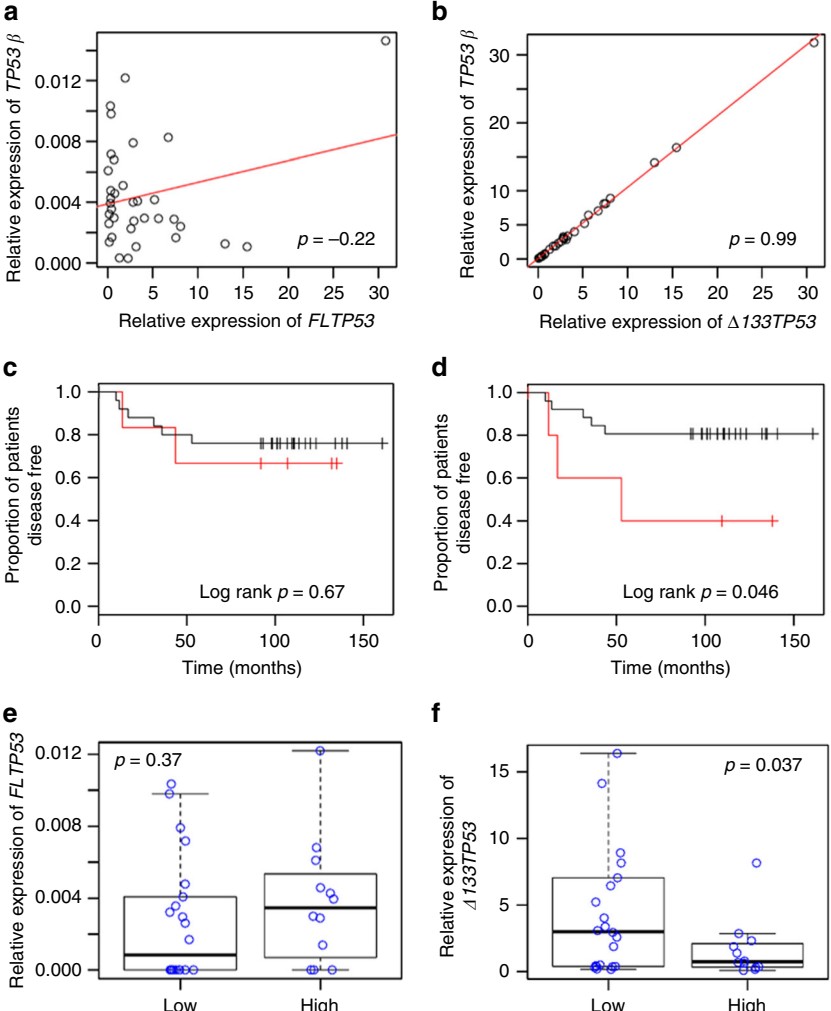

**Fig. 7** Elevated *Δ133TP53* transcript in human colorectal cancer is associated with shorter disease-free survival. **a** Correlation of relative *FLTP53* transcript and *TP53β* transcript over the 35 tumours, $R^2 = -0.22$. **b** Correlation of relative *Δ133TP53* transcript with *TP53β* transcript across the 35 colorectal tumours, $R^2 = 0.99$. **c** Kaplan–Meier curves for 35 colorectal cancer patients illustrating the proportion of disease-free patients stratified by tumours that have a relative *FLTP53* transcript expression <80th percentile (black) vs. those with relative *FLTP53* transcript expression ≥80th percentile (red; log rank test $p = 0.67$). **d** Kaplan–Meier curves for 35 colorectal patients illustrating the proportion of disease-free patients stratified by tumours that have a relative *Δ133TP53* transcript expression <80th percentile (black) vs. those with relative *Δ133TP53* transcript expression ≥80th percentile (red; log rank test $p = 0.046$). **e** Distribution of relative *FLTP53* transcript expression in 34 colorectal tumours stratified by lymphocytic infiltration ($p = 0.37$). **f** Distribution of relative *Δ133TP53* transcript expression in 34 colorectal tumours stratified by lymphocytic infiltration ($p = 0.037$). The line in the middle of each box represents the median, the top and bottom outlines of the box represent the first and third quartiles. $p$-value (**e**, **f**) was obtained after performing a heteroscedastic, two-tailed unpaired $t$-test between different groups. $p < 0.05$ was considered significant

Given this context, we were interested to ask whether IL-6 is an important contributor to the phenotypes of *Δ122p53* mice. To do this we generated *Δ122p53 IL-6* null mice. We found that loss of *IL-6* resulted in reduced levels of multiple serum cytokines and chemokines that are part of the JAK-STAT3 signalling pathway and a reduction in tumour incidence; and where there were sufficient tumours to analyse, mice with loss of IL-6 showed reduced spread to other organs. We also found that IL-6 is required for cell migration, by altering the polarisation of actin fibres, a process driven by RhoA-ROCK signalling, which we found to be upregulated in cells expressing *Δ122p53* or over-expressing *Δ133p53*. Moreover, cell invasion induced by *Δ122p53* or human *Δ133p53* was prevented with inhibitors of JAK-STAT and RhoA-ROCK signalling.

To link the findings from our mouse studies to human malignancies we analysed a cohort of 35 colorectal cancers for expression of *Δ133TP53* mRNA, associated signalling pathways and pathologies. Of interest, we found that the tumours with elevated *Δ133TP53* mRNA were enriched for GPCR signalling upstream of JAK-STAT and RhoA-ROCK consistent with the *Δ122p53* mouse studies. Moreover, these same tumours had significantly lower levels of infiltrating lymphocytes and the corresponding patients had a shorter disease-free survival[50]. There is growing evidence that the presence of immune cells in tumours has a profound effect on tumour progression[51,52] and this is well documented for colorectal cancers where a low 'immunoscore' is associated with poor outcome[37,38]; consistent with our observations. Thus, given our data, it is tempting to speculate that the *Δ133p53* isoform controls the expression of different cytokines dependent on tissue context, to influence the migration of various immune cells in the tumour microenvironment.

Taken together, we propose that some human cancers will have increased levels of the *Δ133p53* isoform. As elevated *Δ133p53* is

only observed in cells with wt *TP53* it seems likely that it is the activation of p53 that transactivates Δ*133TP53* isoform[53,54]. Increased Δ133p53 (by some mechanism) elevates the levels of IL-6 and other pro-inflammatory cytokines. IL-6 is secreted and then binds its receptor to activate the JAK-STAT3 and RhoA-ROCK pathways[28]. This results in activation of the NFκB pathway and the generation of multiple pro-inflammatory chemokines that contribute to migration of the tumour cells, as well as to alterations in the actin cytoskeleton, that promote an invasive phenotype. These cytokines will also contribute to the migration of immune cells into the tumour. This process is outlined in Fig. 8.

Finally, our data suggest that patients whose cancers have elevated Δ133p53 may benefit from treatment with inhibitors of the JAK-STAT and RhoA-ROCK signalling pathways.

## Methods

**Animal cohorts**. Δ*122p53* mice were generated by mating *mΔpro* mice[55] with the *CMV-Cre* recombinase mice to delete the regions between the LoxP sites (exons 3 and 4)[15]. To generate Δ*122p53* and *IL-6* null genotypes, Δ*122/Δ122 IL-6+/+* mice were crossed with *IL-6−/−* mice[17]. Female mice without a functional *p53* allele, suffer significant decreases in embryonic implantation, pregnancy rate and often fail to produce offspring[56]. This is also true for homozygous female Δ*122p53* mice[15]. Therefore, in order to successfully breed all the desired genotype combinations, all female breeders needed to have at least one *wt p53* allele. Additionally, male mice without a *wt p53* allele had a reduced lifespan thus limiting their ability to be productive breeders. Thus, male Δ*122/Δ122 IL-6+/+* were crossed with female *p53+/− IL-6−/−* to generate Δ*122/+ IL-6+/−* mice. Subsequently, these were mated together repeatedly to generate all the genotype combinations, although as few females were generated in this cross (Supplementary Table 2), many breeding pairs were required to ensure sufficient females were obtained. In addition, to increase the numbers of Δ*122/Δ122 IL-6−/−*, a number of male Δ*122/Δ122 IL-6+/−* mice were mated with Δ*122/+ IL-6+/−* mice, which reduced the numbers Δ*122/Δ122 IL-6+/−* for the study. Mouse genotypes were confirmed by PCR using the Kapa Biosystems Mouse Genotyping kit (Catalogue Number KK7302, Kappa Biosystems, USA) on genomic DNA from tail tips for Δ*122p53* and *IL-6* using the following primers. *IL-6* forward primer (FP) (common): 5′-TTCCATCCAGTTGCCTTCTTGG-3′; *IL-6* wild-type reverse primer (RP): 5′-TTCTCATTTCCACGATTTCCCAG-3′; *IL-6* deleted RP: 5′-CCGGA-GAACCTGCGTGCAATCC-3′. Δ*122p53* FP (common): 5′-CAAGTTATGCATC-CATACAG-3′; Δ*122p53* RP: 5′-CCTGCCTCAACTGTCTCTAG-3′; Δ*122p53* deleted RP: 5′-CGTGCACATAACAGACTT-3′. Mice used in these studies were on a C57BL/6 background. All animal research was granted ethical approval by the institutional authorities of participating institutions (University of Otago—AEC 03/12 and Children's Medical Research Institute—AEC C257).

**Animal survival and pathological analyses**. Mice were aged for 600 days and terminated either when visible tumour was apparent, when they lost 20% of body weight, or showed other signs of distress, whichever was earliest. Histological examination was carried out to identify pathology, and tumours were collected and fixed in 10% neutral buffered formalin for at least 24 h followed by standard histological processing and wax embedding. Four micrometre sections were cut followed by hematoxylin and eosin (H&E) staining. Immunohistochemistry analysis was performed on tumour sections to confirm the histopathologic examination. CD3 antibody (#ab5690, 1:400, Abcam, UK) was used to detect T lymphocytes; and B220 (#clone RA3-6B2, 1:100, BD Pharmingen, USA) and paired box transcription 5 (PAX5, # 24/Pax-5, 1:100, BD Pharmingen) antibodies were used to detect B lymphocytes, respectively. All animal experiments were carried out under the respective institutional ethical approval University of Otago (AEC 03/12) and Children's Medical Research Institute (AEC C257).

**Patient samples**. Primary tumour samples were collected from 35 colorectal cancer patients undergoing elective surgery at Dunedin Hospital. A database of clinico-pathological information matched to each of the samples was collected prospectively over a follow-up period of at least 5 years (Supplementary Table 3). These samples were prospectively collected between 1996 and 2007. Informed patient consent was obtained for the use of these tissues and data. Ethical approval had been granted for use of these tissues and data (New Zealand Health & Disability Ethics Committee Reference #14/NTA/33).

**RNA extraction and RT-qPCR**. Total RNA was isolated from snap-frozen tumours as per the manufacturer's protocol using the PureLink RNA mini kit (Invitrogen, USA). RNA (2 µg) extracted from the tumour tissue was reverse transcribed using qScript cDNA SuperMix (Quanta Biosciences, USA), according to the manufacturer's instructions. Primers were designed for specific *TP53* transcript subclasses (*FL*, Δ*133TP53 TP53α* and *β*)[18]. Three reference genes *GAPDH*, *HPRT* and

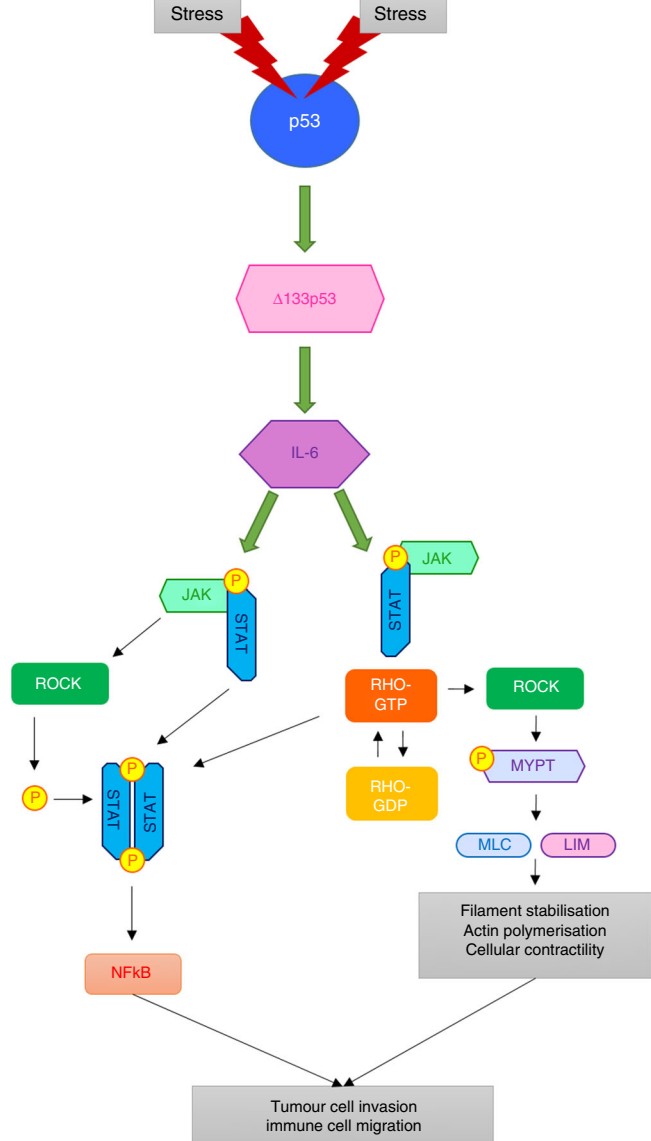

**Fig. 8** Model of how the Δ133p53 isoform regulates the JAK-STAT3 and RhoA-ROCK pathways to promote inflammation and invasion. Wt p53 is activated by some (as yet undefined) stress signal, which turns on Δ133p53. Δ133p53 increases IL-6, which, then binds to its receptor and signals through the JAK-STAT pathway to amplify the IL-6 signal further. The JAK-STAT pathway, now constitutively activated by Δ133p53, drives actin polarisation, cell migration via RhoA-ROCK signalling and alters migration of immune cells

*G6PD* were used. *FLTP53* FP: 5′-CTTCCCTGGATTGGCCA-3′; *FLTP53* RP: 5′-TCTGAAAATGTTTCCTGACTCAGA-3′; Δ*133TP53* FP: 5′- GCCTGAGTGA-CAGAGCAA-3′; Δ*133TP53* RP: 5′- CGTAAGCACCTCCTGCAA-3′; *TP53α* FP: 5′-TTCACCCTTCAGATCCGTG-3′; *TP53α* RP: 5′-GGGCATCCTTGAGTTC-CAA-3′; *TP53β* FP: 5′-TTCAGGACCAGACCAGCTT-3′; *TP53β* RP: 5′-CAAG-TAGCATCTGTATCAGGCAA-3′; *GAPDH* FP: 5′-GAAGGTGAAGGTCGGAGTC-3′; *GAPDH* RP: 5′-GAA-GATGGTGATGGGATTTC-3′; *G6PD* FP: 5′-ATCGACCACTACCTGGGCAA-3′; *G6PD* RP: 5′-TTCTGCATCACGTCCCGGA-3′; *HPRT1* FP: 5′-GAGCACACA-GAGGGCTACAATGT-3′; *HPRT1* RP: 5′-GAAAGGGTGTTTATTCCTCATGGA-3′. RT-qPCR was performed as follows: 100 ng of cDNA was added to 10 µl of SYBR Premix Ex Taq II (Ti RNase H Plus; Takara Bio, Japan) and 200 nM of each primer, in a final volume of 20 µl; the PCR product was run on the Roche LightCycler 480 as follows: 95 °C for 2 min, then 40 cycles of 95 °C for 30 s, 60 °C for 1 min, followed by a melt curve analysis; and RT-qPCR was performed for each sample with each primer pair in triplicate. Relative transcript abundance from RT-

qPCR was calculated using the equation:

$$\text{Amplification efficiency}^{\wedge-(\text{geometric mean reference gene threshold Ct}-TP53\text{ threshold Ct})}$$

**Survival analysis and correlation analysis**. Colorectal tumours were divided into two groups based on the 80th percentile cut-off of the relative expression of either the FLTP53 or Δ133TP53 transcript determined by RT-qPCR. Difference in disease-free survival between the two groups (>80th percentile vs. <80th percentile) of FLTP53 and Δ133TP53 mRNA expression was assessed using Kaplan–Meier analysis followed by log rank test and Cox proportional hazard analyses. This analysis was performed using the survival library in R. A log rank test $p < 0.05$ was considered significant.

**Exon array analyses**. RNA quality extracted from the colorectal tumours was assessed using RNA 6000 NanoChips with the Agilent 2100 Bioanalyzer (Agilent, USA). Samples with a RIN value of >7 were used for the arrays. Exon array hybridisation was performed as per the manufacturer's protocol. In brief, biotin-labelled target was prepared using 1 µg of total RNA from the tumours. Initially, a rRNA removal procedure was performed with the RiboMinus Human/Mouse Transcriptome Isolation Kit (Invitrogen) followed by cDNA synthesis using the GeneChip WT (Whole Transcript) Sense Target Labelling and Control Reagents kit as described by the manufacturer (Affymetrix). The sense cDNA was then fragmented by uracil DNA glycosylase and apurinic/apyrimidic endonuclease 1 and biotin-labelled with terminal deoxynucleotidyl transferase using the GeneChip WT Terminal labelling kit (Affymetrix). Hybridisation was performed using 5 µg of biotinylated target, which was incubated with the GeneChip Human Exon 1.0 ST array (Affymetrix) at 45 °C for 16 h. Following hybridisation, arrays were washed and stained using a Fluidics Station 450 (Affymetrix) and the Affymetrix wash and stain kit following the manufacturer's instructions using Affymetrix fluidics protocol FS450_0001. Nonspecifically bound material was removed by washing and specifically bound target was stained using the anti-streptavidin–phycoerythrin stain provided in the wash and stain kit. The arrays were scanned using the GeneChip Scanner 3000 7 G (Affymetrix) and raw data were extracted from the scanned images and analysed with the Affymetrix Power Tools software package (Affymetrix).

After the microarrays were scanned, quality checks were performed using the Affymetrix Expression Console Software (2012, available from http://www. affymetrix.com/estore/browse/level_seven_software_products_only.jsp? productId=131414&categoryId=35623#1_1), the intensity data were subjected to quantile normalisation and summarisation using the RMA algorithm in the R 'affy'[57] and 'aroma.affymetrix'[58] packages. Analysis was then performed on transcript-level probesets using the core annotations. At the transcript level, 18 707 probesets represented different transcripts.

Δ133TP53 mRNA expression determined by RT-qPCR was combined with the transcript-level Affymetrix exon array data for each CRC. Spearman's correlation was then performed using the cor() function in R with the method set to 'spearman'[59]. Enrichment of gene sets within the input list was determined using the overrepresentation test with default settings using Bonferroni correction in Pantherdb[60].

**Cells and cell lines**. Cells used were as follows: MEFs derived from Δ122/Δ122 IL-6 +/+, Δ122/Δ122 IL–6−/−, Δ122/+ IL-6+/+ and Δ122/+ IL–6−/− mice, respectively; the pancreatic ductal carcinoma (PDAC) cells PDAC/vector and PDAC/Δ122[16]; and the HCT116 human colorectal cancer cells (American Type Culture Collection, USA); PDAC cells were isolated from primary tumours of Pdx1-Cre LSL-Kras$^{G12D/+}$, LSL-Trp53−/− mice[61] and cultured in Dulbecco's modified Eagle medium (DMEM) supplemented with 10% fetal bovine serum (FBS, Moregate, Australia) and 1% penicillin/streptomycin in 20% $O_2$/5% $CO_2$ conditions. Telomerase-immortalised fibroblasts (TIFs) were isolated and cultured in DMEM (Gibco, USA) supplemented with 10% FBS and 1% penicillin/streptomycin in 20% $O_2$/5% $CO_2$ conditions[62]. All cell lines were maintained in DMEM supplemented with 10% FBS in a humidified incubator at 5% $CO_2$ and 37 °C unless specified otherwise.

**Transwell assays**. MEFs were serum-starved in medium with 0.5% FBS for 24 h then harvested, resuspended in serum-deficient medium and seeded into Transwell inserts. After 4 h, cells were fixed using 4% paraformaldehyde, stained with 3% crystal violet, imaged using an Olympus DP71 microscope and analysed using ImageJ software. Data shown are from three biological replicates, each with six technical replicates.

**Cytokine and chemokine analysis**. Serum from 9-week-old Δ122/Δ122 IL-6+/+ ($n = 3$) and 7-week-old Δ122/Δ122 IL–6−/− ($n = 3$) mice showing no visible pathology, were analysed for the expression of cytokines and chemokines using a Bio-Plex Pro Mouse Cytokine 23-plex Assay (Bio-Rad, USA).

**Staining of actin filaments with phalloidin**. Actin filaments were stained using phalloidin. Cells were seeded into a 24-well plate and a scratch wound introduced using a p200 tip when the cells were confluent. Cells were allowed to migrate to close the wound for 4 h, then fixed in 2% paraformaldehyde in phosphate-buffered saline, blocked and permeabilised in 5% bovine serum albumin and 0.2% Triton, then stained with phalloidin-568 (A12380; Alexa Flour; 1:500; Molecular Probes, USA) to label actin filaments and Hoescht 33342 (H1399; 1:1000; Molecular Probes) to label the DNA. Cells were imaged using the Olympus IX71 inverted microscope (Tokyo, Japan) at ×200 magnification and images were merged using Adobe Photoshop. Images of DNA stained with Hoescht were taken first, and then of the filamentous actin stained with phalloidin. This ensured consistency of imaging and no bias towards intensely polarised areas of actin. Quantitation was done with 'blinding' to genotypes by two independent researchers. The amount of actin elongation as a percentage was determined by dividing the number of polarised actin filaments counted by the total nuclei counted. Fifteen images per slide were taken at ×200 magnification, with an average of 20 nuclei per image, making on average 300 nuclei counted for each treatment.

**Organotypic invasion assays**. Invasion of PDAC/Δ122 and PDAC/vector cells through collagen-I matrices was performed[29,30]. Collagen-I was extracted from rat tails and purified to a concentration of 2.5 mg/ml. TIFs were embedded in collagen ($8 \times 10^4$/matrix) and spontaneously remodelled collagen for 12 days in DMEM, supplemented with 10% FBS. Medium was renewed at day 6 of contraction. In all, $4 \times 10^4$ PDAC cells (transduced with either empty vector or Δ122p53) were then seeded on to the contracted matrices, grown for 4 days before being transferred to an air–liquid interface and allowed to invade towards a chemotactic gradient for 12 days. During invasion, PDAC cells were treated with the Pan-JAK inhibitor P6 5 µM[28], the ROCK inhibitors Y-27632 (10 µM) and H1152 (5 µM)[30], and the RhoA inhibitor TAT-C3 (0.25 µg/ml)[29]. Following invasion, samples were fixed in 10% neutral buffered formalin for 24 h and embedded in paraffin for histological analyses[30]. Histological staining was performed on 4 µm sections deparaffinised in xylene and rehydrated using graded ethanol washes. Antigen retrieval was performed for 30 min in a water bath (93 °C for organotypic samples and 100 °C for mouse tissues). For immunohistochemistry staining, samples were quenched in 3% $H_2O_2$ prior to application of protein block (Dako, Agilent). Primary antibodies were incubated (pan-cytokeratin 1:50 C-11 Leica Novocastra, UK; pMYPT1 (Thr696) 1:100 #ABS45 Millipore, USA; pSTAT3(Tyr705) 1:400 #9145 Cell Signaling Technology, USA) and secondary antibodies (Envision, Agilent) coupled to horseradish peroxidase (HRP) were applied. Detection was performed with diaminobenzidine. H&E staining and counterstaining was operated on a Leica autostainer. Invasion index was calculated as per the formula: invasive index = (invasive cells)/(non-invading cells + invasive cells). Data shown are from three technical replicates with three biological replicates each.

**Invasion assays**. Quantification of cell invasion was done in Transwell chambers containing fluorescence-blocking polycarbonate porous membrane inserts (Fluoroblock; #351152; BD Biosciences, USA; pore size 8 µm). A volume of 100 µl of 2 mg/ml Matrigel with reduced growth factors (a commercially prepared reconstituted BM from Englebreth-Holm-Swarm tumours, #354230; BD Biosciences) were prepared in a Transwell. Cells were transfected and treated with Y-27632 (10 µM, Calbiochem, USA) as monolayers before trypsinisation and plating ($1 \times 10^5$) in 2% FBS containing medium on top of a thick layer (around 500 µm) of Matrigel contained within the upper chamber of a Transwell. Controls were left untreated. The upper and lower chambers were then filled with 2% FBS containing media and media with 10% FBS, respectively, thus establishing a gradient of chemo-attractant that permits cell invasion throughout the Matrigel. Cells were allowed to invade at 37 °C, 5% $CO_2$ through the gel before fixing for 15 min in 3.7% formaldehyde. Cells that had invaded through the Matrigel were detected on the lower side of the filter by green fluorescent protein fluorescence and counted. Each assay was performed twice in triplicate for each condition.

**RhoA activity assay**. Cells were lysed in 50 mM Tris, pH 7.2, 1% Triton X-100, 0.5% sodium deoxycholate, 500 mM NaCl, 10 mM MgCl₂, 1 mM phenylmethylsulfonyl fluoride (PMSF) and a cocktail of protease inhibitors. Cleared lysates were incubated with 25 µg of a commercial glutathione S-transferase (GST) fusion protein containing the RhoA-binding domain of Rhotekin-coated beads (GST-RBD, cytoskeleton) for 30 min at 4 °C. Precipitated complexes were washed four times in Tris buffer containing 1% Triton X-100, 150 mM NaCl, 10 mM MgCl₂, 0.1 mM PMSF, eluted in SDS sample buffer, immunoblotted and analysed with antibodies specific for RhoA. Scanned autoradiographs were quantified using Aida/2D densitometry software and normalised to RhoA protein.

**Cell extracts and western blotting**. Medium containing rounded cells was centrifuged at 1200 r.p.m. for 5 min and the pellet containing the rounded invasive cells lysed. The remaining adherent cells were gently scraped in lysis buffer. The two populations of cells were analysed separately. Total protein concentration was determined using the BCA kit (Promega, USA). Protein samples were electrophoretically separated on 8% SDS-PAGE gels and an equal amount of total protein (30 µg) was loaded into each lane. Proteins were transferred onto nitrocellulose

membranes. Membranes were blocked with TBS/0.1% Tween 20 containing 3% milk for 1 h and then incubated overnight with the primary antibodies (mouse anti-ROCK I at 1:400 mouse anti-ROCK II at 1:2500, BD Transduction Laboratories, USA; anti-myc at 1:500 9E10, Thermofisher Scientific, USA) diluted in TBS/ 0.1% Tween 20 containing 3% milk. After several washes in TBS/Tween, membranes were incubated with anti-mouse Ig antibodies linked to HRP. Membranes were developed with ECL according to the manufacturer's instructions. Scanned autoradiographs were quantified using AIDA/2D densitometry software.

**Data availability**. The microarray data set used in the current study is available at the Gene Expression Omnibus under the accession number GSE106535. The authors declare that all remaining data supporting the findings of this study are available within the article and its Supplementary Information file or from the corresponding author upon reasonable request.

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

## Acknowledgements

This work was supported by the Health Research Council of New Zealand, the Royal Society of New Zealand Marsden Fund, the Maurice Wilkins Centre for Molecular BioDiscovery, the Cancer Council NSW and the National Health and Medical Research Council of Australia.

## Author contributions

H.C. wrote the manuscript, performed experiments and analysed data. N.F. performed experiments, analysed and assembled data. I.R. wrote the manuscript, performed experiments and analysed data. S.M. wrote the manuscript, performed experiments and analysed data. A.W. performed experiments, analysed and assembled data. G.W. analysed data. C.V. performed experiments and analysed data. N.A. performed experiments. A.P. performed experiments. F.M. and J.M. provided samples. L.M. performed experiments. M.B. assembled data. M.P., T.L.S., P.T., R.R., P.R., C.P., M.A.B. and A.W.B. are group leaders involved in the project. A.W.B. wrote the manuscript, analysed data and supervised the study together with M.A.B.

## Additional information

**Competing interests:** The authors declare no competing financial interests

