## [Peer Review File · Nature Communications]

Reviewers' comments:

Reviewer #1 Expert in p53 isoforms:

In this manuscript, Campbell et al., demonstrates that D133p53 isoform promotes tumour invasion and metastasis via interleukin-6 activation of JAK-STAT and RhoA/ROCK signalling. The main conclusion is well summarised in the title of the article. The experimental and clinical data supports the conclusion. The finding is totally novel and is of great interest to a broad scientific/clinical community notably with the success and failure of immunotherapy in some cancer patients. This article raises the possibility that D133p53 expression could be used as a biomarker to predict response to immunotherapy.

Some data could/should be strengthened for publication.

Figure 2 and 3. The authors provide the number and type of tumour but not the age at tumour onset. Can the authors perform a statistical analysis to compare the age at cancer onset in the different D122/Il6 mouse models? is it statistically significant? if not, the author should discuss the reasons why?

The figure3A is not easy to read as bars have about the same color and there is no clear separation between the different genotypes. can the authors improve this?

Figure6B and 6D: (western-blot) The authors should show expression of D133 proteins in the western-blot in the corresponding adherent and non-adherent cells.

The figure 7 should be removed because the molecular mechanism of Il6 promoter regulation by D133 and p63/p73 is, in my view, too complex to be addressed in only few experiments in this article. It would require an entire article dedicated to it, which is beyond the scope of this manuscript. The data in figures 1 to 6 are compelling enough to conclude that D133p53 promotes cell invasion via Il6 activation of JAK-STAT and RhoA/ROCK signalling. Therefore only the figure-9c should be presented

Figure8 : Is high expression of D133p53 associated to TP53 mutation? Can the authors discriminate colon cancer according to TP53 mutation or tumour size and then determine whether the patient clinical outcome is associated to D133p53 expression in WT TP53 and/or in mutant TP53 colon cancer and/or tumour size? Is D133p53 expression correlated/associated to inflammatory markers in CRC? Can the authors perform Cox-regression analysis to identify confounding parameter between tumour size, tumour grade, Ras mutation, TP53 mutation, D133p53 expression, Microsatellite instability, Carcinoembryonic antigen, Ki-67, Inflammatory markers, treatment,) ?

Figure8E should be removed.

The authors should discuss their findings in the relation to the review "Emerging cytokine networks in colorectal cancer"

Nat Rev Immunol. 2015 Oct;15(10):615-29. It would strengthen the clinical/biological relevance of their findings.

Reviewer #2 Expert in p53 signalling:

In this study, the authors explore the molecular basis for the oncogenic activity of an isoform of p53, $\Delta 122p53$. Previous studies have shown that mice that expresses $\Delta 122p53$ are tumor prone. Here is argued that this is due to an increase in interleukin 6 expression that mediates the downstream oncogenic events. Data is presented to support a model in which $\Delta 122p53$ binds to

the p53 family members TAp63 and TAp73, thereby sequestering them from occupying the IL6 gene and relieving p63/p73-dependent repression of IL6 expression.

Understanding the role of p53 isoforms in oncogenesis and tumor suppression is a critically important area. Hence, the focus of this manuscript is quite significant. The studies are carefully presented, the data are convincing, and the findings propose to provide new mechanistic insight. Thus, the manuscript is suitable for the readership of Nature Communications. However, several issues need to be addressed before the study should be published.

First, ablation of p63 expression is used to implicate this p53 family member. Similar experiments need to be done with p73 as well. Details concerning the knockdown using shRNA should be included in the manuscript. It is necessary that at least two independent targeting sequences be used and that knockdown efficiency be shown at both the protein and mRNA level to reduce the likelihood of off-target effects. Effects on endogenous IL6 mRNA expression are the key readout.

Second, the authors argue that the effects on IL6 expression are related to gene occupancy by p63 and p73. This needs to be directly shown using chromatin immunoprecipitation assays that demonstrate that expression of $\Delta 133p53$ blocks the interaction of p63 and p73 with the IL6 promoter.

Third, the interaction of $\Delta 133p53$ and p63 and p73 should be demonstrated with endogenous proteins.

Finally, the authors should discuss the details of the generation of $\Delta 122p53$ mouse model in this manuscript, so that the reader does not have to refer back to previously published studies.

Reviewer #3 Expert in p53 signalling:

Remarks to the Author:

In this study, Campbell and co-authors suggest that $\Delta 133p53$ promotes tumor invasion and metastasis via interleukin-6 activation of JAK-STAT and RhoA-ROCK signaling. The authors first crossed $\Delta 122p53$ mice (equivalent to human $\Delta 133p53$), previously shown to induce lymphomagenesis, with IL-6^{-/-} mice to generate a series of strains containing one or both copies of $\Delta 122p53$ together with wt, IL-6^{+/-} or IL-6^{-/-} alleles, respectively. They show that deletion of IL-6 reduced the tumorigenesis of the $\Delta 122p53$ mice. Cell culture models together with the use of JAK-STAT3 and RhoA-ROCK inhibitors suggest that $\Delta 122p53$ -induced tumorigenesis is driven by IL-6 overexpression. They further show that $\Delta 122p53$ binds to TAp63 and TAp73 at IL-6 promoter to relieve their inhibition of IL-6 transcription.

The finding is interesting and the data is well presented. However, the conclusion of IL-6 as a tumorigenesis-driven cytokine is somewhat not firmly supported without proper controls and most of the data are correlative. Given that $\Delta 122p53$ promotes tumorigenesis and increases IL-6 have been previously shown, the current finding is considered to be additive. There is also no direct evidence supporting that $\Delta 122p53$ inhibits p63 and p73 at IL6 gene promoters.

Concerns:

1. The authors need to provide at least some rationale as to why they specifically study IL-6. How about other cytokines?

2. Fig. 1 and other experiments: Wild-type and IL6^{-/-} control mice are needed to justify the conclusion. The reduction of these cytokines and tumorigenesis (below) by deleting IL6 is not

surprising, as it promotes tumorigenesis by regulating all hallmarks of cancer and microenvironment, including survival, proliferation, invasiveness and metastasis as well as metabolism.

3. Is the difference of tumor incidence between $\Delta 122/\Delta 122$; IL-6+/+ mice and $\Delta 122/\Delta 122$; IL-6-/- mice statistically significant? Tumor incidence between $\Delta 122/+$; IL-6+/+ mice and $\Delta 122/+$; IL-6-/- mice? The numbers (n) in the Fig. 2 is confusing. They should be the total number of mice, but the pies represent the mice with tumors. Again, is there any statistic significance for tumor types between different strains? All the statistic analysis is important, considering that there is no difference of animal survival (Fig. S4). The authors indicate that the lack of significance might be due to animals being sacrificed because of distress unrelated to malignancy. This should be explained. For example, how many mice in each group were taken down and when they were taken down. Does this affect the results of tumor incidence? As most of the mice developed tumor, how do you justify that the distress unrelated to tumors contributes to this statistic insignificance? The authors indicated that the animals were monitored for up to 600 days. However in Fig. S4b, there are still animals beyond 600 days, please explain.

4. Line 433. Still, the conclusion "IL-6 is the driver of the pro-inflammatory and metastatic phenotypes of $\Delta 122p53$ mice" needs to be justified, given that IL-6 is critical for tumorigenesis and inflammation response and there is lack of animal survival and statistic analysis of tumor incidence, latency and tumor types. How about crossing IL6 -/- mice with other cancer models? What if deleting other cytokines reported in previous paper upregulated by $\Delta 122p53$? Proper negative controls are needed to make the strong (namely "driver") conclusion.

5. Fig. 3: statistic analysis is needed.

6. line 445, why pancreatic cancer model was used here? Why not lymphoma or sarcoma model observed in animal models (Fig. 2)? The cell lines used in other assays are also mixed. For example, colon rectal cancer cell lines were used in Fig. 6. H1299 cells were used in Fig. 7F.

7. Fig. 6C: how much percentage of rounding cells (GFP-positive) is non-adherent? Fig. 6E, a representative immunoblot to demonstrate GTP-bound RhoA should be shown.

8. Line 514: ME180 cells (GDS2534) and $\Delta 122p53$ splenocytes (GSE27586) are different cells. Are the results comparable?

9. Figs. 7D and 7E: Anti-p53 antibodies or control isotype-matched IgG should be using for IP to demonstrate the IPed bands are specific. What about IL-6 expression upon knockdown of p63 or p73? What is the mechanisms underlying the $\Delta 122p53$ binding to p63 and p73 resulting upregulation of IL6?

10. Line 563: Fig. 6E should be Fig. 8E. The correlation of elevated $\Delta 133TP53$ with reduced TP63 and TP73 in tumors seems to be confusing. If previous figures show that 133p53 function to binds to p63/p73 and suppresses their function, it indicates a selective pressure for 133p53 high cells for high expression of p63/p73. If p63/p73 are already low in these tumors, why do you need $\Delta 133TP53$ to suppress their function?

11. Discussion needs to be more insightful, not just summarizing the results

12. Is IL6 upregulated in your microarray analysis listed in supple table 2?

Reviewer 1 Expert in p53 isoforms:

Query: *Figure 2 and 3. The authors provide the number and type of tumour but not the age at tumour onset. Can the authors perform a statistical analysis to compare the age at cancer onset in the different D122/Il6 mouse models? is it statistically significant? if not, the author should discuss the reasons why?*

Response: We do not have "age of tumour onset" data which requires live imaging equipment which is not available; we only have "endpoint" data (*i.e.* the time when the mice were culled) which is provided in Supplementary Figure 5A. Statistical analysis shows that there is no significance difference in the time at which tumours were detected between the genotypes (Supplementary Figure 5B). A comment about this has been added to the text (highlight on p15).

Note: We are not sure that these extra data add value to the paper, but they are included for now for reviewer 1 to decide whether the data should be included or not.

Query: *The figure3A is not easy to read as bars have about the same color and there is no clear separation between the different genotypes. can the authors improve this?*

Response: To make Figure 3A clear, we have replaced the bar graph with two tables detailing the comparison between the genotypes using the chi-square statistic and calculated p value (Figure 3C and Figure 3D). Figure 3C provides a comparison of T- lymphomas with local spread versus those that had extensive spread and multiple organ involvement. $\Delta 122/\Delta 122$ IL6+/+ had significantly more organ involvement and distant spread than $\Delta 122/\Delta 122$ IL6-/- mice ($p=0.01$). Figure 3D provides a comparison of metastatic versus non-metastatic sarcomas for heterozygous $\Delta 122/+$ mice. Those that were IL6+/+ had significantly more metastases than IL6+/- mice ($p=0.03$). We have added the statistical analyses to the text (highlights on pp.13,14).

Query: *Figure6B and 6D: (western-blot) The authors should show expression of D133 proteins in the western-blot in the corresponding adherent and non-adherent cells.*

Response: These data have now been included in Figure 6B and 6D respectively.

Query: *The figure 7 should be removed because the molecular mechanism of Il6 promoter regulation by D133 and p63/p73 is, in my view, too complex to be addressed in only few experiments in this article. It would require an entire article dedicated to it, which is beyond the scope of this manuscript. The data in figures 1 to 6 are compelling enough to conclude that D133p53 promotes cell invasion via Il6 activation of JAK-STAT and RhoA/ROCK signalling. Therefore only the figure-9c should be presented.*

Response: We thank the reviewer for his/her advice and agree that the regulation of the IL-6 promoter by $\Delta 133$ and p63/p73 is complex and beyond the scope of this manuscript. Based on the reviewer's advice, we have removed Figure 7, 8E and 9C from the manuscript.

Query: *Figure8 : Is high expression of D133p53 associated to TP53 mutation? Can the authors discriminate colon cancer according to TP53 mutation or tumour size and then determine*

whether the patient clinical outcome is associated to D133p53 expression in WT TP53 and/or in mutant TP53 colon cancer and/or tumour size? Is D133p53 expression correlated/associated to inflammatory markers in CRC? Can the authors perform Cox-regression analysis to identify confounding parameter between tumour size, tumour grade, Ras mutation, TP53 mutation, D133p53 expression, Microsatellite instability, Carcinoembryonic antigen, Ki-67, Inflammatory markers, treatment,)? Figure 8E should be removed.

Response: We have carried out Sanger sequencing of the *TP53* gene (from exons 5- 8) for tumours expressing high $\Delta 133TP53$ (>80th percentile). Our results show that all tumours with high $\Delta 133TP53$ expression have wild type *TP53*. A statement to this effect has been added to the text (highlight at the top of p18). As advised by the reviewer we have looked at stage, grade, and microsatellite instability, however, as we have only 35 tumours in this study, we don't have statistical power to perform Cox-regression analysis to identify confounding parameters. We have however included lymphocyte infiltration data, which is an indicator of poor prognosis, and this is highly correlated with $\Delta 133TP53$ expression but not *FLTP53* (Figure 7E and F), confirming the predictive value of $\Delta 133TP53$. As advised by the reviewer we have removed Figure 8E from the manuscript. All data relating to the tumour analysis are now in Figure 7. These data are described on p18 (highlighted paragraph).

Query: *the authors should discuss their findings in the relation to the review "Emerging cytokine networks in colorectal cancer" Nat Rev Immuno. 2015 Oct; 15(10): 615-629. It would strengthen the clinical/biological relevance of their finding.*

Response: A paragraph discussing the role of the isoform in regulating the immune cell environment in CRCs has been added to the Discussion (see 2nd paragraph on p20). We have also extensively rewritten the Discussion to discuss p53 and isoforms in the context of immune regulation.

Reviewer #2 Expert in p53 signalling:

Query: *First, ablation of p63 expression is used to implicate this p53 family member. Similar experiments need to be done with p73 as well. Details concerning the knockdown using shRNA should be included in the manuscript. It is necessary that at least two independent targeting sequences be used and that knockdown efficiency be shown at both the protein and mRNA level to reduce the likelihood of off-target effects. Effects on endogenous IL6 mRNA expression are the key readout.*

Second, the authors argue that the effects on IL6 expression are related to gene occupancy by p63 and p73. This needs to be directly shown using chromatin immunoprecipitation assays that demonstrate that expression of $\Delta 133p53$ blocks the interaction of p63 and p73 with the IL6 promoter.

Third, the interaction of $\Delta 133p53$ and p63 and p73 should be demonstrated with endogenous proteins.

Response: We acknowledge the shortcomings of the data in Figure 7 and have therefore taken the advice of reviewer 1 and removed the figure.

Query: *Finally, the authors should discuss the details of the generation of $\Delta 122p53$ mouse model in this manuscript, so that the reader does not have to refer back to previously published studies.*

Response: We have now included the details of the generation of the $\Delta 122p53$ mouse and the various $\Delta 122p53$ IL-6 mutants in the Materials and Methods section of the manuscript (see highlighted paragraph on pp 4,5).

Reviewer #3 Expert in p53 signalling:

Query: 1. The authors need to provide at least some rationale as to why they specifically study IL-6. How about other cytokines?

Response: IL-6 was 12 fold higher in the serum of $\Delta 122p53$ mice compared to wtp53 mice; much higher than other pro-inflammatory cytokines we measured. We have included a rationale to this end in the text on page 12 (see highlight).

Query: 2. Fig. 1 and other experiments: Wild-type and IL6^{-/-} control mice are needed to justify the conclusion. The reduction of these cytokines and tumorigenesis (below) by deleting IL6 is not surprising, as it promotes tumorigenesis by regulating all hallmarks of cancer and microenvironment, including survival, proliferation, invasiveness and metastasis as well as metabolism.

Response: As requested by the reviewer we have provided the cytokine analyses from the serum of p53^{+/+} IL-6^{-/-} mice in a modified Figure 1 and Supplementary Figure S1. Overall sera from wtp53/IL-6^{-/-} control mice generally showed similar levels of these molecules to $\Delta 122/\Delta 122$ IL-6^{+/+} mice or were lower (e.g, Tnf- α Ccl-2, and IL-17 ($p < 0.05$)) as we found previously. A description of these data has been added (see highlight on p12).

Query 3: A. Is the difference of tumor incidence between $\Delta 122/\Delta 122$; IL-6^{+/+} mice and $\Delta 122/\Delta 122$; IL-6^{-/-} mice statistically significant? Tumor incidence between $\Delta 122/\Delta 122$; IL-6^{+/+} mice and $\Delta 122/\Delta 122$; IL-6^{-/-} mice? **B.** The numbers (n) in the Fig. 2 is confusing. They should be the total number of mice, but the pies represent the mice with tumors. **C.** Again, is there any statistic significance for tumor types between different strains? All the statistic analysis is important, considering that there is no difference of animal survival (Fig. S4). **D.** The authors indicate that the lack of significance might be due to animals being sacrificed because of distress unrelated to malignancy. This should be explained. For example, how many mice in each group were taken down and when they were taken down. Does this affect the results of tumor incidence? As most of the mice developed tumor, how do you justify that the distress unrelated to tumors contributes to this statistic insignificance? **E.** The authors indicated that the animals were monitored for up to 600 days. However in Fig. S4b, there are still animals beyond 600 days, please explain.

Response:

Response to part A: To address this we have included a table summarising the chi-square statistics comparing the incidence of tumours between the different genotypes. This is shown in Figure 2G. $\Delta 122/\Delta 122$ IL6^{+/+} mice had significantly more tumours than $\Delta 122/\Delta 122$ IL6^{-/-} mice ($p=0.036$), but not with the IL6^{+/-} mice due to low numbers. Heterozygous $\Delta 122/+$ IL6^{+/+} mice had significantly more tumours than those that had lost one ($p=0.01$) or both ($p=0.015$) IL6 alleles. We have added the statistical analyses to the text (highlights on pp.13,14).

Response to part B: Sorry if there is some confusion, but the data in Figure 2 A-F are the total number of mice in the study, not just the mice with tumours. The pies provide details on type and incidence of tumours in the individual genotypes.

Response to part C: There is no statistical difference in the tumour type after loss of IL-6 on either the $\Delta 122/\Delta 122$ or the $\Delta 122/+$ background.

Response to parts C and D: We have now replaced the survival curves that have been censored at 600 days. The curves now include animals with and without malignancy (Supplementary Figures S4A and S4B). There is still no significant difference between genotypes. On reflection, the explanation we provided in the earlier version of this paper is wrong. The lack of significance is best explained by there being too few animals with no malignancy, which limits statistical power. We are grateful to the reviewer for requiring us to scrutinize our data with more care.

In addition, we have included the survival data for the $p53+/+ IL6-/-$ cohort which shows that loss of IL6 alone does not predispose to malignancy (they were culled due to a distended abdomen and one mouse was culled at day 58 due to a rectal prolapse) and $p53+/+ IL6+/+$ controls (see Supplementary Figure 4A). The addition of these extra data address the "other experiments" comment under Query 2 (first line). The text has been altered on p15 to reflect the altered survival analysis and inclusion of extra data.

Query 4: A. Line 433. Still, the conclusion "IL-6 is the driver of the pro-inflammatory and metastatic phenotypes of $\Delta 122p53$ mice" needs to be justified, given that IL-6 is critical for tumorigenesis and inflammation response and there is lack of animal survival and statistic analysis of tumor incidence, latency and tumor types. **B.** How about crossing IL6 -/- mice with other cancer models? What if deleting other cytokines reported in previous paper upregulated by $\Delta 122p53$? **C.** Proper negative controls are needed to make the strong (namely "driver") conclusion.

Response:

Response to part A: We have modified this statement and it now reads as: "These results suggest that IL-6 is an important contributor to the pro-inflammatory and tumorigenic phenotypes of $\Delta 122p53$ mice." (see highlight on p15).

Response to part B: It would be interesting to cross IL-6-/- mice with other cancer models. Our goal however is to define the oncogenic mechanism of $\Delta 122p53/\Delta 133p53$. We could delete other cytokine genes but IL6 was most elevated in the $\Delta 122p53$ mice, so it seemed the best candidate to investigate.

Response to part C: We have removed the term "driver" from the text in various places.

Query 5. Fig. 3: statistic analysis is needed.

Response: As suggested by the reviewer we have carried out statistical analyses and summarized these results in a table format in Figure 3C and Figure 3D.

Query: 6. line 445, why pancreatic cancer model was used here? Why not lymphoma or sarcoma model observed in animal models (Fig. 2)? The cell lines used in other assays are also mixed. For example, colon rectal cancer cell lines were used in Fig. 6. H1299 cells were used in Fig. 7F.

Response: The organotypic invasion assays using mouse pancreatic cancers is a well-established model and we had previously shown that $\Delta 122p53$ could promote invasion in this system. These cells thus seemed like the most appropriate model to test the effect of JAK-STAT inhibitors. Similarly, HCT-116 cells have been widely used to study invasion in a quantitative manner (eg PMID 17275810, 3039715). Again, they seemed like a good model to use. Lymphoma cells cannot be used in these assays as they grow in suspension.

Upon the recommendation of reviewer 1 we have removed Figure 7 from this manuscript and hence the H1299 cell line is no longer part of this manuscript.

Query: 7. Fig. 6C: how much percentage of rounding cells (GFP-positive) is non-adherent? Fig. 6E, a representative immunoblot to demonstrate GTP-bound RhoA should be shown.

Response: As requested we have provided the information on the percentage of non-adherent GFP-positive cells in the legend to Figure 6. "The total number of transfected (GFP positive) cells was $46 \pm 5.8\%$, of which $12.6 \pm 4.4\%$ were non-adherent." We have also included a representative immunoblot demonstrating GTP-bound RhoA along with the quantitation as Figure 6E as requested.

Query: 8. Line 514: ME180 cells (GDS2534) and $\Delta 122p53$ splenocytes (GSE27586) are different cells. Are the results comparable?

Response: As we have removed Figure 7 upon the recommendation of reviewer 1, the bioinformatic data using ME180 cells in the previous Supplementary Figure 6 are no longer relevant and have been removed from the paper.

9. Figs. 7D and 7E: Anti-p53 antibodies or control isotype-matched IgG should be using for IP to demonstrate the IPed bands are specific. What about IL-6 expression upon knockdown of p63 or p73? What is the mechanisms underlying the $\Delta 122p53$ binding to p63 and p73 resulting upregulation of IL6?

10. Line 563: Fig. 6E should be Fig. 8E. The correlation of elevated $\Delta 133TP53$ with reduced TP63 and TP73 in tumors seems to be confusing. If previous figures show that 133p53 function to binds to p63/p73 and suppresses their function, it indicates a selective pressure for 133p53 high cells for high expression of p63/p73. If p63/p73 are already low in these tumors, why do you need $\Delta 133TP53$ to suppress their function?

Response to Queries 9 and 10: As stated above, we acknowledge the limitations of the data in Figure 7 and have taken the advice of reviewer 1 and removed the Figure.

Query: 11. Discussion needs to be more insightful, not just summarizing the results

Response: We have extensively rewritten the Discussion to discuss the role of p53 and its isoforms in regulating genes involved in inflammation and other aspects of the immune response (see highlights on pp 18-20). We have also included a paragraph about immune cell infiltration in colorectal cancers, prognosis and a role for the $\Delta 133p53$ isoform as requested by reviewer 1 (2nd paragraph on p20 - also highlighted).

Query: 12. *Is IL6 upregulated in your microarray analysis listed in supple table 2?*

Response: At the mRNA level, IL-6 levels do not correlate with *Δ133TP53* mRNA expression, suggesting that this association is post-transcriptional. It appears as noted by reviewer 1, that the regulation of IL6 is complex.

REVIEWERS' COMMENTS:

Reviewer #1 (Remarks to the Author):

It is not known why some patients will respond well to immunotherapy or targeted therapy while others patients will respond poorly or even suffer cancer progression. It is well established that p53 regulates multiple components of the immune response, making it a pivotal player in the response to immunotherapy, targeted therapy, chemotherapy, radiotherapy, endocrine therapy, However, regardless of its mutation status, p53 can both activate and inhibit the same components of the immune response. The seemingly contradictory regulations of the immune response by p53 could be due to the differential expression of the p53 isoforms that is different depending on the cell types. The p53 isoforms would explain the difficulties in the clinic to predict response to immunotherapy and other anti-cancer therapies

In previous studies, the authors and other labs have independently shown that the D133p53 isoforms (alpha and beta) promote cancer cell invasion and endothelial cell migration (angiogenesis) by inducing cytokines expression and secretion. However, it is not known which cytokines are active. The authors have previously reported that $\Delta 122p53$ mice (a model of human $\Delta 133p53$ isoform) are tumour prone, have extensive inflammation and 12-fold more serum IL-6 than wild-type mice, suggesting that IL-6 play a role in $\Delta 122p53$ -mediated invasive activity. To determine whether IL6 mediates $\Delta 122p53$ -dependent inflammation and metastasis formation, the authors crossed $\Delta 122p53$ mice with IL-6 knock-out mice. To corroborate the genetic evidence of the $\Delta 122p53/IL6^{-/-}$ mouse model and to assess the relevance to human biology, the authors have performed in-vitro experiments in mouse (MEF, PDAC) and human cell lines (colorectal cancer HCT116). They have also completed the study by conducting a clinical analysis in a cohort of colorectal cancer patients. They found that patients with elevated $\Delta 133p53$ splice variant levels have a shorter disease free survival, supporting the authors' conclusions.

The authors have taken account of all referees' comments, which has strengthened the data and conclusions. The figure-1 and figure-4C are striking. All the data together clearly demonstrates that IL-6 mediates D133/D122-promoted invasion and metastasis formation by activating the JAK-STAT3 and RhoA/rock pathways.

The manuscript shows the biological importance of the complex interplay between the p53 isoforms and IL6, which are involved in many different biological processes. The manuscript should be of interest to a wide audience and will be at the origin of many studies.

Reviewer #2 (Remarks to the Author):

The authors have addressed the concerns of the previous review and the manuscript is now acceptable for publication.

Reviewer #3 (Remarks to the Author):

The removal of Figure slightly reduces my enthusiasm on this study.

Reviewer #1 (Remarks to the Author):

It is not known why some patients will respond well to immunotherapy or targeted therapy while others patients will respond poorly or even suffer cancer progression. It is well established that p53 regulates multiple components of the immune response, making it a pivotal player in the response to immunotherapy, targeted therapy, chemotherapy, radiotherapy, endocrine therapy, However, regardless of its mutation status, p53 can both activate and inhibit the same components of the immune response. The seemingly contradictory regulations of the immune response by p53 could be due to the differential expression of the p53 isoforms that is different depending on the cell types. The p53 isoforms would explain the difficulties in the clinic to predict response to immunotherapy and other anti-cancer therapies

In previous studies, the authors and other labs have independently shown that the D133p53 isoforms (alpha and beta) promote cancer cell invasion and endothelial cell migration (angiogenesis) by inducing cytokines expression and secretion. However, it is not known which cytokines are active. The authors have previously reported that $\Delta 122$ p53 mice (a model of human $\Delta 133$ p53 isoform) are tumour prone, have extensive inflammation and 12-fold more serum IL-6 than wild-type mice, suggesting that IL-6 play a role in D122p53-mediated invasive activity. To determine whether IL6 mediates D122p53-dependent inflammation and metastasis formation, the authors crossed $\Delta 122$ p53 mice with IL-6 knock-out mice. To corroborate the genetic evidence of the $\Delta 122$ p53/IL6^{-/-} mouse model and to assess the relevance to human biology, the authors have performed in-vitro experiments in mouse (MEF, PDAC) and human cell lines (colorectal cancer HCT116). They have also completed the study by conducting a

clinical analysis in a cohort of colorectal cancer patients. They found that patients with elevated $\Delta 133$ p53 splice variant levels have a shorter disease free survival, supporting the authors' conclusions.

The authors have taken account of all referees' comments, which has strengthened the data and conclusions. The figure-1 and figure-4C are striking. All the data together clearly demonstrates that IL-6 mediates D133/D122-promoted invasion and metastasis formation by activating the JAK-STAT3 and RhoA/rock pathways.

The manuscript shows the biological importance of the complex interplay between the p53 isoforms and IL6, which are involved in many different biological processes. The manuscript should be of interest to a wide audience and will be at the origin of many studies.

Response: We thank the reviewer for complementary comments about the revised manuscript. There are no issues to address.

Reviewer #2 (Remarks to the Author):

The authors have addressed the concerns of the previous review and the manuscript is now acceptable for publication.

Response: We thank the reviewer for considering our revised manuscript acceptable for publication. There are no issues to address.

Reviewer #3 (Remarks to the Author):

The removal of Figure slightly reduces my enthusiasm on this study.

Response: Removal of the figure concerning the interaction of the D133p53 isoform with p63 and p73 was recommended by reviewer 1 as it was considered too complex and beyond the scope of our paper - and in fact not necessary, as the rest of the paper was compelling enough. We took the advice of reviewer 1.

Reviewer 3 does not raise any issue needing to be addressed.

Thanking you,

Antony Braithwaite